# Conserved stromal–immune cell circuits secure B cell homeostasis and function

Mechthild Lütge ●[1], Angelina De Martin[1], Cristina Gil-Cruz[1], Christian Perez-Shibayama[1], Yves Stanossek[1,2], Lucas Onder[1], Hung-Wei Cheng ●[1], Lisa Kurz[1], Nadine Cadosch[1], Charlotte Soneson[3], Mark D. Robinson ●[3], Sandro J. Stoeckli[2], Burkhard Ludewig ●[1,4]✉ & Natalia B. Pikor ●[1,4]✉

B cell zone reticular cells (BRCs) form stable microenvironments that direct efficient humoral immunity with B cell priming and memory maintenance being orchestrated across lymphoid organs. However, a comprehensive understanding of systemic humoral immunity is hampered by the lack of knowledge of global BRC sustenance, function and major pathways controlling BRC–immune cell interactions. Here we dissected the BRC landscape and immune cell interactome in human and murine lymphoid organs. In addition to the major BRC subsets underpinning the follicle, including follicular dendritic cells, *PI16*[+] RCs were present across organs and species. As well as BRC-produced niche factors, immune cell-driven BRC differentiation and activation programs governed the convergence of shared BRC subsets, overwriting tissue-specific gene signatures. Our data reveal that a canonical set of immune cell-provided cues enforce bidirectional signaling programs that sustain functional BRC niches across lymphoid organs and species, thereby securing efficient humoral immunity.

Tissue homeostasis is defined as the regulated state of a tissue where supportive cells form optimized microenvironments to fulfill a tissue's primary functions[1]. In the context of secondary lymphoid organs (SLOs), fibroblastic reticular cells (FRCs) form immune-supportive niches that are sustained throughout the steady state and during adaptive immune cell priming[2–4]. Within the B cell follicle (BF), *Cxcl13*-expressing B cell zone RCs (BRCs) steer B cell, antigen and leukocyte encounters thereby increasing germinal center (GC) efficiency[5]. In contrast to the elaborate experimental toolbox to assess B cell biology, means to study FRC immunobiology are scarce, leaving BRC function almost entirely inferred from B cell perturbation studies[6–12]. As such, little is known about the functional overlap between BRC subsets or the signals sustaining BRC subset identity.

SLOs consist of lymph nodes (LNs), Peyer's patches and splenic white pulp (WP), which differ in their anatomy, developmental origin and surveyed bodily fluids[13–15]. Distinct SLOs share key BRC subsets such as follicular dendritic cells (FDCs), while subsets supporting memory B cells (MBCs) in the antigen-sampling zone[16] or plasma cells in the medulla[17] are not ubiquitous across SLOs. The cross-organ overlap, identity and function of BRCs has not been examined, leaving it unclear whether functionally analogous BRC subsets support circulating lymphocytes. Resolving such key questions pertaining to BRC identity and function is pivotal to understanding the role of fibroblastic niches in systemic humoral immunity.

Recent studies began to elucidate tissue-specific fibroblast heterogeneity, revealing overlapping and tissue-specific or disease-specific features[18,19]. Such global assessments of transcriptomic diversity of fibroblasts across lymphoid and nonlymphoid tissues[19,20] overlook the contribution of tissue identity and inflammatory state on specialized

[1]Institute of Immunobiology, Kantonsspital St.Gallen, St. Gallen, Switzerland. [2]Department of Otorhinolaryngology Head and Neck Surgery, Kantonsspital St.Gallen, St. Gallen, Switzerland. [3]Department of Molecular Life Sciences and Swiss Institute of Bioinformatics, University of Zurich, Zurich, Switzerland. [4]These authors contributed equally: Burkhard Ludewig, Natalia B. Pikor. ✉e-mail: burkhard.ludewig@kssg.ch; natalia.pikor@kssg.ch

FRC function. In particular, low-abundance and difficult-to-isolate RCs, such as FDCs, are vastly underrepresented. In this study, we used the Cxcl13-Cre/TdTomato R26R-enhanced yellow fluorescent protein (EYFP) (abbreviated to Cxcl13-Cre/TdTomato EYFP) mouse model[5,21] to enrich for CXCL13+ cells from LNs, splenic WP and Peyer's patches. While we define the shared composition of marginal RCs (MRCs), FDCs, T–B border RCs (TBRCs) and a subset of *Pi16*+ RCs across SLOs, we found no overlapping niche factors among SLO-specific BRC subsets. Cross-organ transcriptomic analyses of BRCs and immune cells revealed that circulating immune cells instructed the functional convergence of shared BRC subsets that expressed a canonical set of niche factors. These bidirectional signals were recapitulated across murine SLOs and validated in human LN and tonsillar tissues. Our study reveals conserved feedforward circuits that sustain homeostatic BRC niches, ultimately securing efficient humoral immunity within and across SLOs.

## Results

### Heterogeneity of the BRC landscape across murine SLOs

Whole and cross-tissue transcriptomic analyses comparing distinct stromal cell types often fail to capture underrepresented subsets such as FDCs[15]. In this study, we used the Cxcl13-Cre/TdTomato EYFP mouse model to define the molecular landscape of BRCs. Microscopy analysis of LNs, spleen and Peyer's patches highlighted the colocalization of transgene-targeted cells that are marked by expression of the red fluorescent protein TdTomato[5,21] to BFs across SLOs (Fig. 1a–c and Extended Data Fig. 1a–c). Higher-resolution microscopy and flow cytometry analyses revealed tissue-specific BRC features. In LNs, TdTomato+ networks extended from LYVE1+ lymphatics (Fig. 1d). TdTomato+ cells were almost entirely non-endothelial cells expressing podoplanin (PDPN) although with varying degrees of bone marrow stromal cell antigen-1 (CD157) and stem cell antigen-1 (SCA1) expression (Fig. 1e,f and Extended Data Fig. 1d). In the spleen, TdTomato+ BRCs underpinned WP but not marginal zone B cells (Fig. 1g). A small percentage of transgene-targeted cells were endothelial cells, although most splenic TdTomato+ cells lacked CD31 and PDPN expression but expressed CD157 (Fig. 1h,i and Extended Data Fig. 1e). In Peyer's patches, TdTomato+ BRCs extended from the subepithelial antigen-sampling zone through the BF (Fig. 1j). TdTomato+ cells were primarily non-endothelial, PDPN+ cells (Fig. 1k,l and Extended Data Fig. 1f). Further delineation of CD157+ cells revealed the consistent demarcation of BF BRCs across SLOs (Extended Data Fig. 1g–i). Collectively, TdTomato+ cells identified follicular BRCs and extra-follicular cells across SLOs, with organ-specific differences in the relative abundance and patterning of BRC networks.

### BRC and *Pi16*+ RC subset identity across SLOs

To better resolve the BRC landscape within and across SLOs, we performed single-cell RNA sequencing (scRNA-seq) analyses of Cxcl13-Cre/TdTomato+ cells sorted from the spleen, LNs and Peyer's patches of naive mice and from the LNs and spleens of vesicular stomatitis virus (VSV)-infected mice (Fig. 2a). To ensure organ-specific cluster assignment[15], we first characterized TdTomato+ cells from each SLO individually, detecting nine subsets in the LNs and spleen, and seven subsets in Peyer's patches (Extended Data Fig. 2a–f). Three subsets consistently showed a high expression of *Cxcl13* light zone (LZ) FDCs that expressed high levels of *Cr2* and *Sox9* (ref. [22])), dark zone (DZ) FDCs with lower expression of FDC marker genes and *Cxcl12* transcript expression[5], and MRCs that coexpressed *Madcam1* and *Tnfsf11*. In all SLOs we detected TBRCs that expressed *Ccl19*, *Ccl21a* and *Fmod* and a subset of *Pi16*-expressing RCs (*Pi16*+ RCs). Additionally, we observed organ-specific RCs with low *Cxcl13* expression that could be characterized based on published marker genes. LN-specific BRC subsets included medullary RCs (MedRCs), interfollicular RCs, *Cd34*-expressing perivascular RCs (PRCs) and a subset of *Cxcl9*-expressing TBRCs[5,22,23] (Extended Data Fig. 2a,b). Spleen-specific subsets included red pulp

(RP) fibroblasts, capsular and bridging channel RCs (BCRCs), as well as a PRC subset[24,25] (Extended Data Fig. 2c,d). Subepithelial and lamina propria *Cxcl13*-expressing fibroblasts were unique to the intestinal tissue surrounding Peyer's patches[26] (Extended Data Fig. 2e,f). Integration of *Cxcl13*+ cells across SLOs revealed that subsets underpinning the BF and *Pi16* RCs clustered together, whereas SLO-specific BRC subsets clustered separately, suggesting that shared BRC subsets expressed convergent molecular identities in addition to canonical marker genes (Fig. 2b,c).

The topological distribution of shared BRC subsets was confirmed by confocal microscopy. MRCs consistently underpinned the antigen-sampling zone, FDCs constituted a sub-MRC network and TBRCs were located at the T–B borders (Fig. 2d–f). Co-staining for PI16 and CXCL13 protein or TdTomato was technically not feasible, hence leaving the exact localization of *Pi16*+ RCs among the *Cxcl13*-expressing cells elusive. Despite the conservation of canonical gene signatures (Fig. 2g), shared BRC subsets still segregated by SLO when re-embedded (Fig. 2h,i). Thus, while known marker gene expression and topology were conserved across SLOs, shared BRC subsets maintained tissue-specific gene signatures.

### Developmental and anatomical gene sets imprint BRC identity

To delineate SLO-specific gene signatures independent of anatomically unique subsets, we ran a comparative analysis of shared BRC subsets only. Pseudobulk-level multidimensional scaling (MDS) revealed the main clustering of samples by organ but no segregation of samples from immunized or naive mice (Fig. 3a). Variance partition analysis confirmed that most variance in gene expression among BRCs could be explained by SLO and subset identity, but not by activation status (Extended Data Fig. 3a). This was consistent with differential gene expression analysis that revealed only moderate transcriptional changes following VSV immunization in both LNs and spleen (Extended Data Fig. 3b,c). In contrast, enrichment analysis of SLO-specific gene signatures revealed tissue-specific gene sets that reflected developmental programs, including the organ-specific expression of key transcription factors and transcriptional regulators (Fig. 3b,c). Splenic BRCs expressed *Nkx2-5*, *Nkx2-3* and *Tlx1*, which have a role in splenic WP development[27]. Homeobox genes imprinting lower thoracic and lumbar region development were expressed by inguinal LN (ILN) (*Hoxd8*, *Hoxc9*, *Hoxd9*) and Peyer's patch (*Hoxc8*) BRCs[28]. LN RCs exhibited a *Meox2* gene signature and Peyer's patch BRCs expressed *Sfrp1* implicated in intestinal Wnt signaling[29]. Additionally, tissue-specific gene sets reflected distinct organ anatomy, including iron ion homeostasis in the spleen[30] and retinoic acid metabolism in Peyer's patches[31]. Thus, developmental and anatomical programs imprinted SLO-specific signatures among shared BRC subsets.

### Bidirectional cues define BRC–immune cell interaction

Despite bearing organ-specific gene signatures, we reasoned that shared BRC subsets probably expressed more than a handful of canonical marker genes defining their transcriptional convergence (Fig. 2b). Visualization of BRCs using diffusion maps as a nonlinear dimensionality reduction technique[32] confirmed the ordering of BRCs according to subset and SLO identity in the first and second dimensions, respectively (Fig. 4a,b). Gene set enrichment analysis (GSEA) identified highly specific pathways for each BRC subset (Fig. 4c), which consisted of unique sets of niche factors (Fig. 4d), suggesting the potential to engage diverse leukocyte populations. GSEA also identified BRC-intrinsic signaling pathways that were conserved across SLOs. FDCs expressed genes involved in NF-κB signaling, while MRCs demonstrated a response to fibroblast growth factor (FGF). TBRCs demonstrated a response to type I interferon (IFN) and *Pi16*+ RCs expressed genes associated with a response to transforming growth factor β (TGFβ) (Fig. 4e,f). Consistently, FDCs showed an increased expression of genes involved

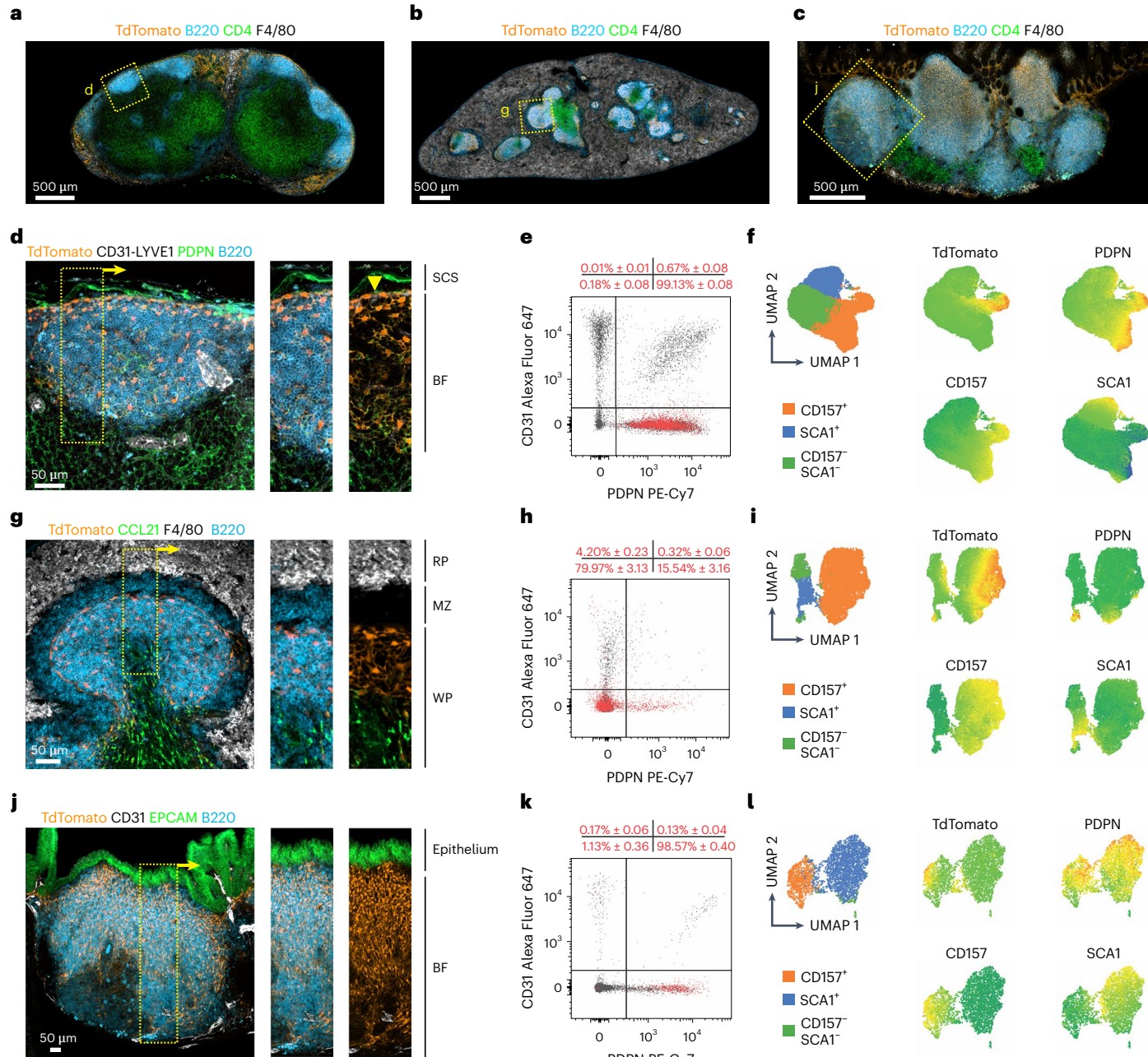

**Fig. 1 | Anatomical heterogeneity of Cxcl13-Cre⁺ BRC landscapes across SLOs. a–c**, Representative immunofluorescence images of ILNs (**a**), spleen (**b**) and Peyer's patches (**c**) from naive Cxc13-Cre/TdTomato EYFP mice. Immunostaining for B220, CD4 and F4/80 demarcated leukocyte compartments and TdTomato identified Cxcl13-Cre transgene expression. **d**, Representative immunofluorescence images of the BRC network in an ILN primary BF, with the SCS, BF and lymphatics (arrowhead) demarcated. **e,f**, Flow cytometry analysis of non-hematopoietic Cxcl13-Cre/TdTomato-targeted cells in the ILNs of naive Cxcl13-Cre/TdTomato EYFP mice (**e**) and relative expression of TdTomato, PDPN, CD157 and SCA1 by TdTomato⁺ cells (**f**). **g**, Representative immunofluorescence images of the BRC network in the splenic WP BF, with the RP, marginal zone (MZ)

and WP regions indicated. **h,i**, Flow cytometry analysis of non-hematopoietic Cxcl13-Cre/TdTomato-targeted cells in the spleens of naive Cxcl13-Cre/TdTomato EYFP mice (**h**) and the relative expression of TdTomato, PDPN, CD157 and SCA1 by TdTomato⁺ cells (**i**). **j**, Representative immunofluorescence images of the BRC network in Peyer's patches with the epithelium and BF demarcated. **k,l**, Flow cytometry analysis of non-hematopoietic Cxcl13-Cre/TdTomato-targeted cells in Peyer's patches of naive Cxcl13-Cre/TdTomato EYFP mice (**k**) and the relative expression of TdTomato, PDPN, CD157 and SCA1 by TdTomato⁺ cells (**l**). In **a–c,d,g,j** data are representative of at least six mice. In **e,f,h,i,k,l** n = 6 from two independent experiments. In **e,h,k** TdTomato⁺ BRC subsets were quantified according to CD31 and PDPN expression; data are shown as the mean ± s.e.m.

in the interleukin-4 (IL-4) pathway that signals via NF-κB, while *Pi16*⁺ RCs expressed genes involved in canonical and noncanonical TGFβ signaling (Extended Data Fig. 4a).

To assess to what extent these signaling programs were suggestive of BRC modulation by immune cells, we performed scRNA-seq on hematopoietic cells from the LNs and spleens of VSV-infected mice,

and Peyer's patches from uninfected mice (Fig. 4g and Extended Data Fig. 4b). In contrast to BRCs, leukocytes did not segregate by SLO, suggesting that their transcriptome was shaped by cell type identity but not by current tissue residence (Fig. 4h,i). After cluster characterization (Extended Data Fig. 4c), naive B cells, *Bcl6*-expressing GC B cells and follicular helper T (T_FH) cells, *Prdm1*⁺ plasma cells, macrophages and

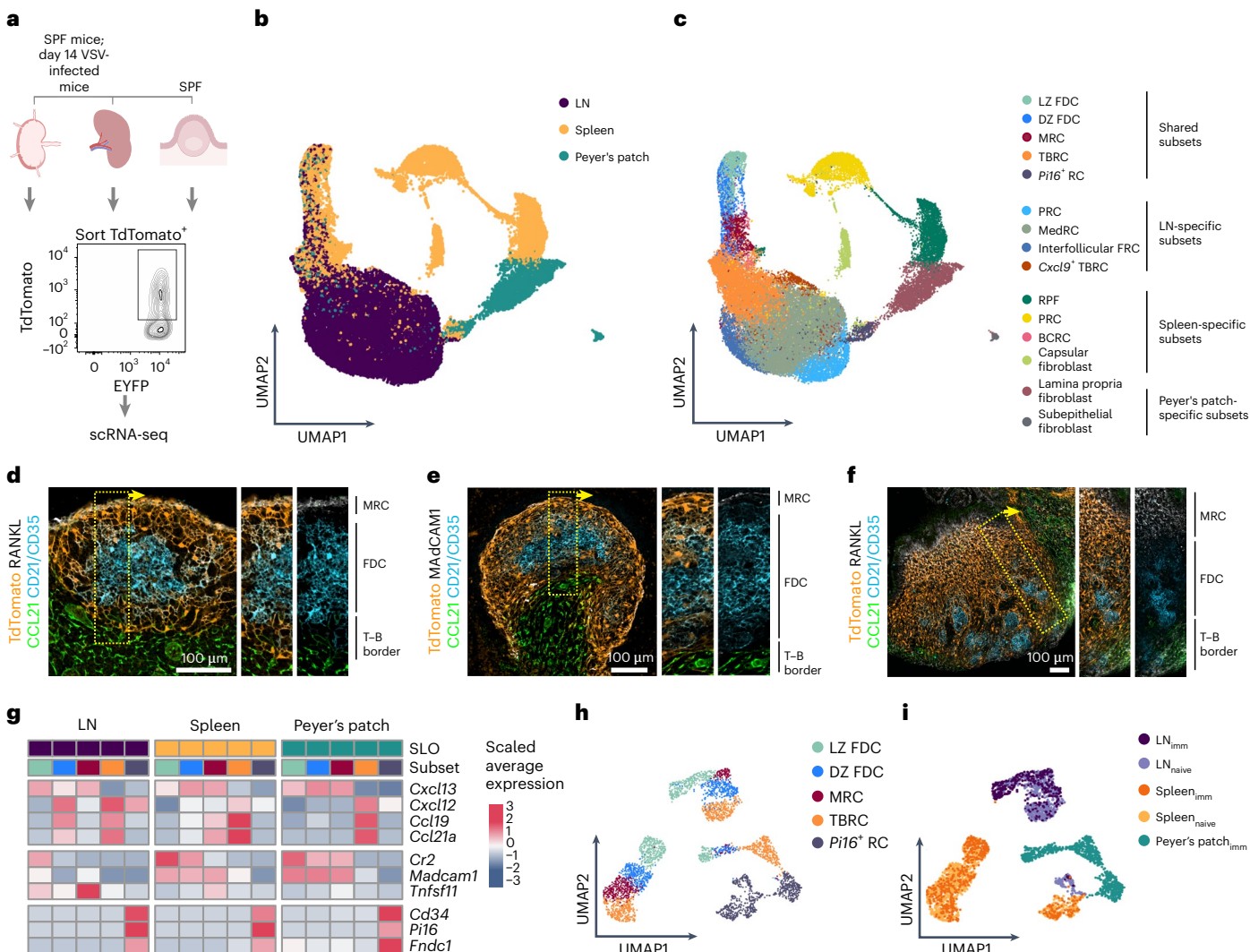

**Fig. 2 | BRC and *Pi16*+ RC subsets are shared across SLOs. a**, Schematic representation of TdTomato+ cell isolation from LNs, spleen and Peyer's patches from either specific pathogen-free (SPF) mice or day 14 VSV-infected mice for scRNA-seq. **b,c**, Uniform manifold approximation and projections (UMAPs) visualizing *Cxcl13*-expressing RCs from ILNs, spleen and Peyer's patches from naive and VSV-immunized Cxcl13-Cre/TdTomato mice after integration across SLOs and colored by SLO (**b**) or assigned subset identity (**c**). **d–f**, Immunostaining of the BFs from ILNs (**d**), spleen (**e**) and Peyer's patches (**f**) from Cxcl13-Cre/TdTomato mice. Immunostaining with RANKL or MAdCAM1 demarcates MRCs, CD21/CD35 demarcates FDCs and CCL21 demarcates the T–B border region. **g**, Heatmap showing the average expression of signature genes across shared

BRC subsets. **h,i**, UMAP visualization of re-embedded BRC subsets that are shared across SLOs downsampled to a maximum of 300 cells per subset of each SLO and colored by subset identity (**h**) and SLO (**i**). In **b,c** the scRNA-seq data represent 34,538 *Cxcl13*+ cells from eight independent experiments with *n* = 7 biological replicates for LNs, *n* = 7 biological replicates for the spleen and *n* = 5 biological replicates for Peyer's patches. In **d–f** images are representative of at least six mice per staining and organ. In **g–i** scRNA-seq data represent 9,572 *Cxcl13*+ cells (downsampled to 3,361 *Cxcl13*+ cells for visualization) from eight independent experiments with *n* = 7 biological replicates for LNs, *n* = 7 biological replicates for the spleen and *n* = 5 biological replicates for Peyer's patches.

---

DCs were used to predict subset-specific BRC interactions (Fig. 4j). The anticipated receptor–ligand expression of *Ltbr-Lta-Ltbr-Ltb* and *Tnfrsf1b-Tnf* confirmed known mediators of FRC differentiation[15]. Key BRC–immune cell interactions conserved across SLOs encompassed genes coding for chemokine receptor–ligand pairs, integrins and selectins (*Icam1-Lfa1*, *Cd34-Sell*), immune complex presentation and sampling (*Cr2-Fcer2a*), immune cell activation and survival (*Il6-Il6ra*, *Il7-Il7r*, *Tnfsf13b-Tnfrsf13c*), immune synapse formation (*Plxnb1-Sema4d*) and Notch signaling (*Dll1/4-Notch1*) (Fig. 4j and Extended Data Fig. 4d). Consistent with GSEA, MRCs encoding *Fgfr2* were predicted to interact with myeloid cells; *Pi16*+ RCs expressed transcripts encoding for TGFβ receptors were predicted to interact with ligands expressed by macrophages and B cells. FDCs were predicted to

interact with *Il4*-expressing T$_{FH}$ cells, *Il1b*-expressing myeloid cells and *Vegfb*-expressing GC B cells. The expression pattern of several receptor–ligand pairs was highly specific for a given BRC subset and immune cell type across SLOs, which is suggestive of conserved BRC–immune cell interactions.

## Leukocyte ligands specify BRC differentiation and activation

To determine whether predicted BRC–immune cell interactions were topologically feasible, we performed spatial transcriptomic analyses of murine LNs using the 10X Genomics Visium platform. Although this platform captures up to ten cells per spot, we reasoned that because all immune cell-provided, BRC specification cues were soluble, they could be effective within a 27.5-μm radius that is captured by one

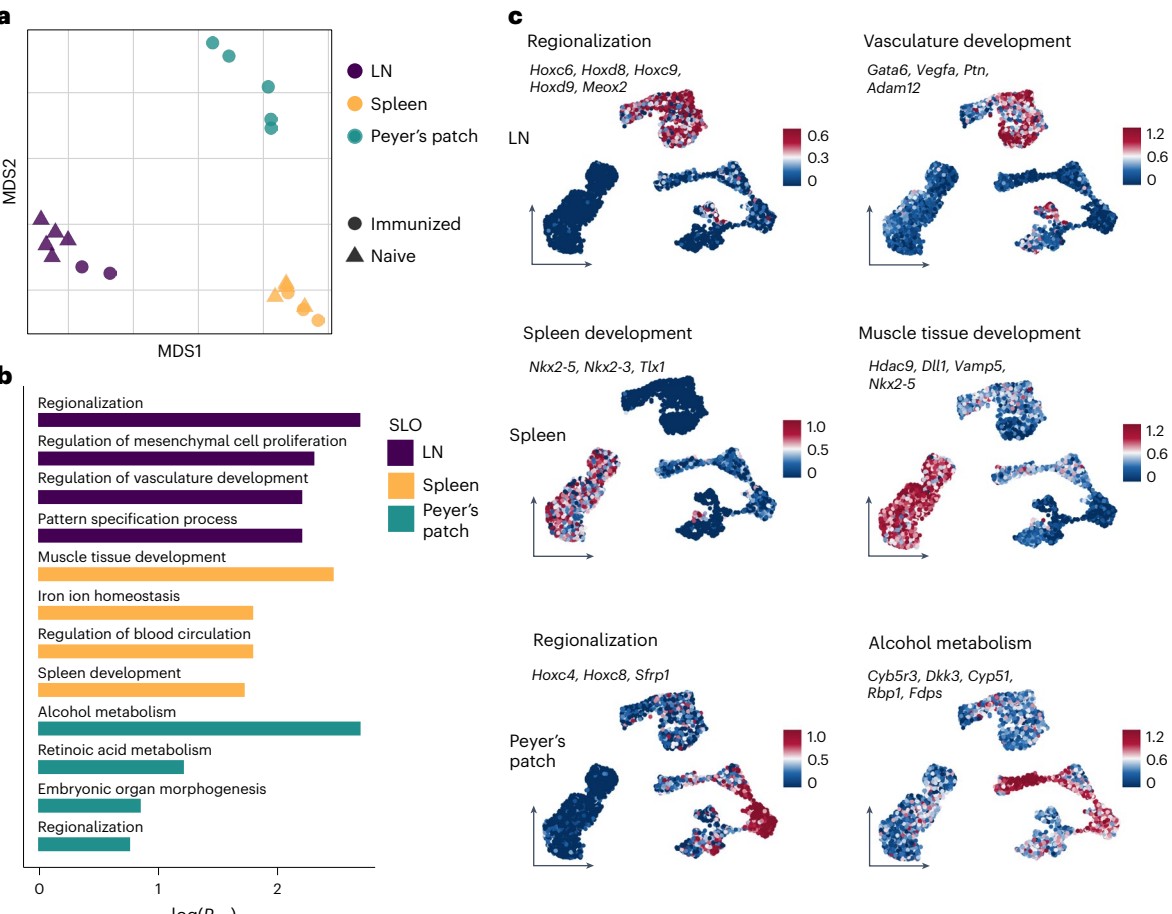

**Fig. 3 | Developmental and anatomical genes imprint BRC identity.**
**a**, Pseudobulk-level MDS plot for the visualization of sample distances colored by SLO and shaped by condition (naive or VSV-immunized). **b**, SLO-specific Gene Ontology (GO) terms as determined by an enrichment test based on SLO-specific genes across shared BRC subsets. **c**, SLO-specific gene signatures derived from enriched pathways and projected as average gene expression onto the UMAP

of shared BRC subsets. In **b** $P$ adjustment was performed using the Benjamini–Hochberg procedure. In **a**–**c** scRNA-seq data represent 9,572 $Cxcl13^+$ cells (downsampled to 3,361 $Cxcl13^+$ cells for visualization) from eight independent experiments with $n = 7$ biological replicates for LNs, $n = 7$ biological replicates for the spleen and $n = 5$ biological replicates for Peyer's patches.

spot. Immunofluorescence staining of B cells confirmed that $Cxcl13$ expression underpinned areas of B cell accumulation (Fig. 5a,b and Extended Data Fig. 5a). Cell type decomposition mapping of the abundance and distribution of BRCs and immune cells revealed that MRC and LZ FDC profiles were primarily expressed toward the apex and center of BFs, respectively, while DZ FDC and TBRC transcriptomes were enriched at the follicle boundary toward the T cell zone (TZ) (Fig. 5c). $Pi16^+$ RC transcriptomes were located at the subcapsular and interfollicular regions and the medulla, all areas with abundant B cell aggregation. Transcriptional profiles from B cells and $T_{FH}$ cells were detected within BFs, while macrophages mapped more broadly across LN regions. Based on deconvoluted cell type compositions, we quantified the colocalization of each BRC subset and immune cell type as a relative fraction of shared spots. These data indicated that naive B cells, plasma cells and macrophages colocalized to a similar extent with all BRC subsets; GC B cells were predicted to be preferentially situated in the neighborhood of follicular BRC subsets and, to a lower extent, $Pi16^+$ RCs; $T_{FH}$ cells were predicted to preferentially colocalize with LZ FDCs and $Pi16^+$ RCs (Fig. 5d). Furthermore, projections of signaling pathways related to NF-κB signaling and chondrocyte differentiation were enriched in the vicinity of FDCs; gene sets related to the response to TGFβ and regulation of angiogenesis mapped to $Pi16^+$ RC-underpinned regions, which is suggestive of subset-specific

BRC responsiveness to factors produced by colocalized immune cells (Extended Data Fig. 5b).

Next, we assessed the biological function of immune cell-provided factors on BRC differentiation and activation. Interactome analysis identified IL-4, interleukin-1β (IL-1β) and vascular endothelial growth factor B (VEGF-B) as FDC interacting factors, and TGFβ1 and progranulin (PRGN) as $Pi16^+$ RC stimuli (Fig. 4j and Extended Data Fig. 4d). To assess the impact of candidate stimulators on BRC identity, LN fibroblasts from Cxcl13-Cre/TdTomato EYFP mice were cultured for 48 h with the respective ligands. TGFβ1, PRGN, IL-1β and VEGF-B, but not IL-4, induced $Cxcl13$ transcription as indicated by the TdTomato fluorescence on EYFP$^+$CD45$^-$CD31$^-$ cells[5] (Fig. 5e,f). These data indicate that immune cell-provided factors sustain BRC identity.

We next sought to evaluate the biological function of predicted stimuli. Stimulation of LN fibroblasts with TGFβ1 and PRGN resulted in attenuated production of the $Pi16^+$ RC niche factor, interleukin-6 (IL-6) (Fig. 5g). Subcutaneous injection of recombinant murine IL-1β, IL-4 and VEGF-B identified IL-1β as a factor that promotes FDC expansion in LNs, while IL-4 caused an increase in the expression of the FDC niche factor ICAM1 and enhanced the frequency and extent to which FDCs capture phycoerythrin–immune complexes (PE–ICs) (Fig. 5h–l and Extended Data Fig. 6a–c). Consistent with the

expression of Fc receptors on B cells and of LFA-1 on $T_{FH}$ cells (Fig. 4j), the cytokine-steered maturation of FDCs probably influences antigen sampling and $T_{FH}$ cell function[33] (Fig. 5m). Collectively, the conserved expression of receptor–ligand pairs and the observed biological effects on BRCs suggest that TGFβ1 and PRGN influence IL-6 production by $Pi16^{+}$ RCs, while IL-1β and IL-4 specify FDC differentiation and activation, respectively (Fig. 5m).

### BRCs steer $T_{FH}$ cell differentiation via IL-6

In addition to immune cell-derived factors promoting subset specification, BRCs produced several niche factors that steer leukocyte migration and survival[5,9,11,12,34]. Among BRC-produced cytokines, interactome analyses predicted $Pi16^{+}$ RCs to express the highest transcript levels of $Il6$ and $T_{FH}$ cells to be the main recipients of IL-6 signals (Fig. 4j). To evaluate the function of BRC-produced IL-6 on GC responses, we immunized Cxcl13-Cre/TdTomato $Il6^{loxP/loxP}$ mice and littermate controls with VSV and examined responses in draining ILNs 14 d later. The selective deletion of $Il6$ in Cxcl13-Cre/TdTomato-targeted cells had no impact on the frequency of $T_{FH}$ cells (Fig. 6a,b). However, consistent with the role of IL-6 in suppressing regulatory T ($T_{reg}$) cells[35], IL-6 attenuated the relative proportion of FOXP3$^{+}$ follicular regulatory T ($T_{FR}$) cells[36,37] (Fig. 6c,d). Together with an increased proportion of $T_{FR}$ cells, Cxcl13-Cre/TdTomato $Il6^{loxP/loxP}$ mice displayed a reduced frequency of GC B cells and attenuated neutralizing antibody titers (Fig. 6e–g). Thus, production of IL-6 by Cxcl13-Cre/TdTomato-targeted cells sustains GC responses through downtuning of $T_{FR}$ cells.

### BRC–immune cell circuits are conserved across species

To examine to what extent BRC molecular identity was conserved across species, we performed scRNA-seq analyses of non-hematopoietic cells from human LNs and tonsils (Fig. 7a, Extended Data Fig. 7a,b and Extended Data Table 1). $CXCL13^{+}$ BRC subsets in human LNs consisted of FDCs, TBRCs and MRCs, while tonsils consisted of FDCs, TBRCs and $PI16^{+}$ RCs but not MRCs (Fig. 7b). In human LNs, we did not identify a sufficient number of $PI16^{+}$ RCs, probably given the relative scarcity of $PI16^{+}$ RCs coexpressing $CXCL13$. Human BRCs expressed similar canonical marker gene signatures as their murine counterparts (Fig. 7c). Although we detected one FDC cluster, we observed a partitioning in the expression of the LZ FDC markers $CR2$ and $FCER2$ in one half of the FDC cluster and $CXCL12$ expression in the opposing portion of the cluster, recapitulating LZ and DZ phenotypes in human FDCs (Extended Data Fig. 7c). Correlation analysis based on the average expression of homologs from the most variable genes confirmed a strong conservation of BRC subset identities across species (Fig. 7d). Visualization of BRC networks in human LNs and tonsils revealed that BFs were underpinned by PDPN$^{+}$ cells, with clusterin (CLU)-expressing FDCs at the center of the follicle (Fig. 7e,f). Perivascular cells expressed high amounts of ACTA2 (smooth muscle actin), while T cell zone RCs (TRCs) expressed low levels of ACTA2, which was absent from the BF. CXCL13 was expressed by MRCs in human LNs (arrows) and by PDPN$^{+}$ FDCs in the tonsils. CCL19$^{+}$ cells predominantly lined the T–B border and the TZ but were also observed in human LN MRCs concomitant with CD3$^{+}$ cells accumulation (Fig. 7e,f, arrowheads).

As in mice, variance partition analysis confirmed tissue origin, but also subset and patient identity as drivers of variance (Extended Data Fig. 7d,e). Tissue specificity was imprinted by developmental and anatomical programs driven by unique sets of transcription factors in human BRCs (Extended Data Fig. 7f,g). Nevertheless, analysis of subset-specific gene set enrichment revealed that human FDCs expressed genes involved in leukocyte proliferation, MRCs showed an enrichment for mononuclear cell migration, leukocyte chemotaxis genes were enriched in TBRCs and $PI16^{+}$ RCs expressed genes related to cell–substrate adhesion (Fig. 7g). Examination of pathway genes confirmed the expression of homologous niche factors by BRCs across species (Fig. 7h and Fig. 4d). GSEA additionally revealed the cross-species conservation of subset-specific, BRC-intrinsic signaling programs. FDC transcriptomes were enriched for genes involved in NF-κB signaling and chondrocyte differentiation, while $CXLC13$-expressing $PI16^{+}$ RCs expressed genes associated with response to TGFβ (Fig. 7i).

To assess whether immune cell-provided BRC specification cues were conserved across species, we performed scRNA-seq analyses on immune cells from the LNs and tonsils of patients (Extended Data Fig. 8a,b and Extended Data Table 1). Immune cells clustered by cell type but not tissue of origin (Extended Data Fig. 8c,d). Based on marker gene expression, we defined subsets of naive cells and MBCs, GC B cells, plasma cells, $T_{FH}$ cells, macrophages and DCs (Extended Data Fig. 8d,e). Consistent with the expression of functionally homologous BRC niche factors (Fig. 7g,h) and signaling pathways (Fig. 7i), a marked overlap in the expression patterns of receptor–ligand pairs between mouse and human was observed (Fig. 7j and Extended Data Fig. 9a). BRC-provided niche factors conserved in human BRCs included FDC-specific provision of ICAM2 and complement receptors, and $PI16^{+}$ RC-provided IL-6. In turn, human FDCs were predicted to receive IL-1β and IL-4 from their interacting immune cell partners, and $PI16^{+}$ RCs were predicted to respond to leukocyte-produced TGFβ and PRGN. To evaluate whether immune cell-derived cues affected human FRC differentiation, tonsillar fibroblasts were treated in vitro with the predicted FDC and $PI16^{+}$ RC ligands. Stimulation with IL-1β induced the expression of the FDC-associated transcription factor $Sox9$ (ref. 22), while TGFβ1 was observed to induce $PI16$ expression in bulk fibroblast tissue cultures (Extended Data Fig. 9b,c) and sorted $PI16^{+}$ RCs (Extended Data Fig. 9d). Thus, human BRCs exhibited a marked cross-species topological and molecular conservation encompassing a core set of bidirectional niche factors and specification cues.

## Discussion

BRCs have emerged as a tripartite player in the control of B cell responses, steering antigen distribution and availability, T–B cell interaction and affinity maturation. Nevertheless, the molecular pathways specifying organ and species identity as well as the homeostatic control of BRC–immune cell niches are unknown, precluding a comprehensive understanding of systemic humoral immunity. Our study shows that BRC identity is dichotomously shaped by SLO identity and immune cell-provided specification cues. Regionalized developmental programs, organ anatomy and the metabolic milieu imprint tissue-specific gene signatures. Despite the uniform GC output of MBCs and plasma

**Fig. 4 | Conserved molecular circuits define BRC–immune cell interaction topology. a,b**, Diffusion map of BRC subsets that are shared across SLOs colored by subset identity (**a**) and split by SLO (**b**). **c**, Conserved subset-specific niche factors derived from significantly enriched GO terms determined by enrichment test on subset-specific genes. **d**, Average gene expression of subset-specific niche factors derived from enriched pathways and projected onto the diffusion map. **e**, Conserved subset-specific signaling pathways derived from significantly enriched GO terms determined by enrichment tests of subset-specific genes. **f**, Subset-specific enriched pathways projected as the average gene expression of the indicated genes onto the diffusion map. **g**, Schematic representation of cell isolation and processing of immune cells for interactome analysis. **h,i**, UMAP visualization of immune cells from ILNs, spleen and Peyer's patches from VSV-immunized Cxcl13-Cre/TdTomato mice colored by subset identity (**h**) or SLO (**i**). **j**, Heatmap with average gene expression of conserved receptor–ligand pairs as determined by CellPhoneDB analysis, reflecting niche factors provided by BRC or immune cell-provided BRC maturation cues. In **c,e** $P$ adjustment was performed using the Benjamini–Hochberg procedure. In **a–f** scRNA-seq data represent 9,572 $Cxcl13^{+}$ cells from eight independent experiments with $n = 7$ biological replicates for LNs, $n = 7$ biological replicates for the spleen and $n = 5$ biological replicates for Peyer's patches. In **h,i** scRNA-seq data represent 23,250 immune cells from one independent experiment with $n = 4$ individual mice.

cells across SLOs, no transcriptional overlap was determined between tissue-specific BRC niches such as MedRCs. Uniquely BRCs and *Pl16*⁺ RCs exhibited a functional convergence, probably driven by the

reiterated interaction with defined leukocyte populations. In support of this, we unveiled an evolutionary conservation in bidirectional cues exchanged between BRCs and leukocytes, inferring a feedforward

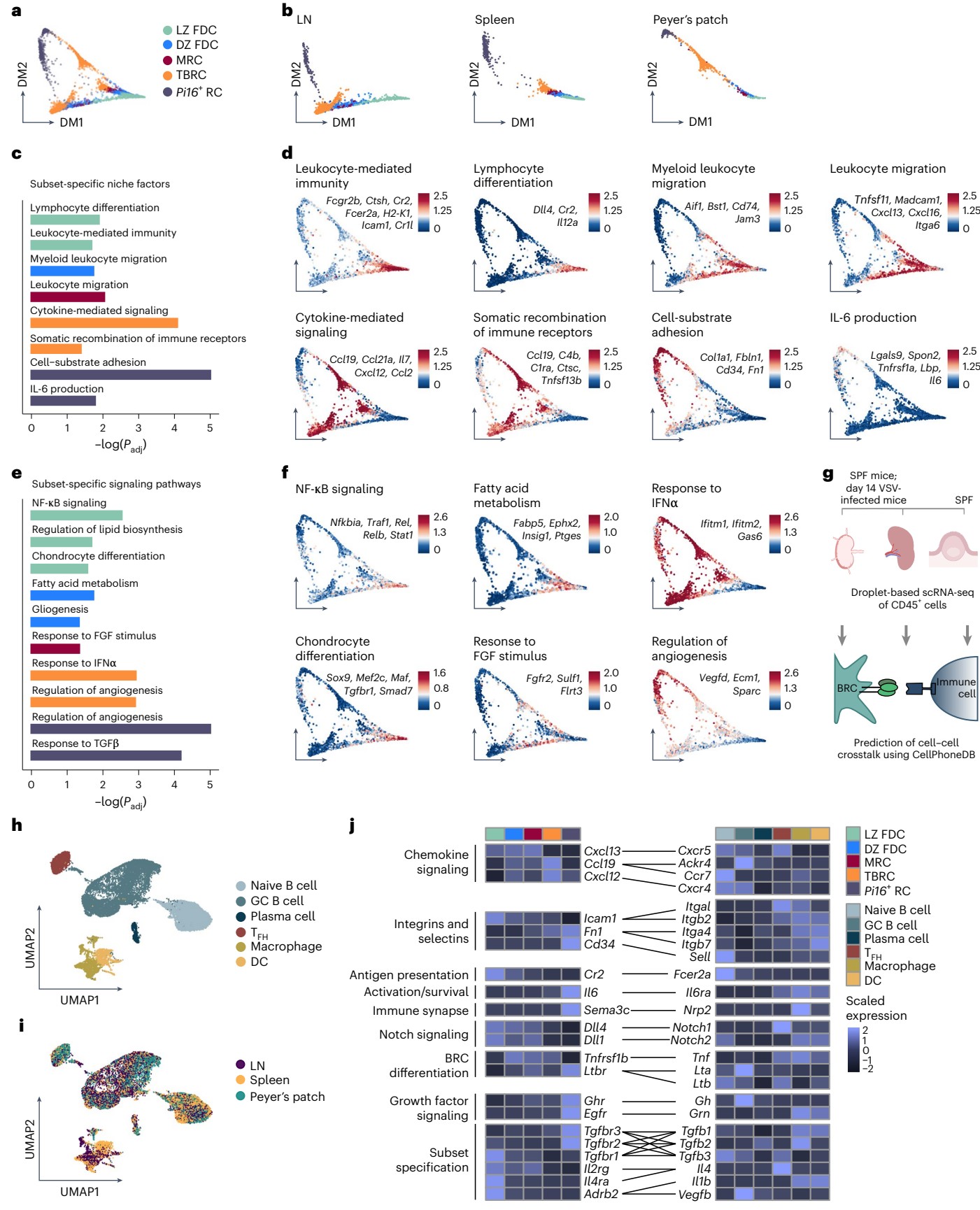

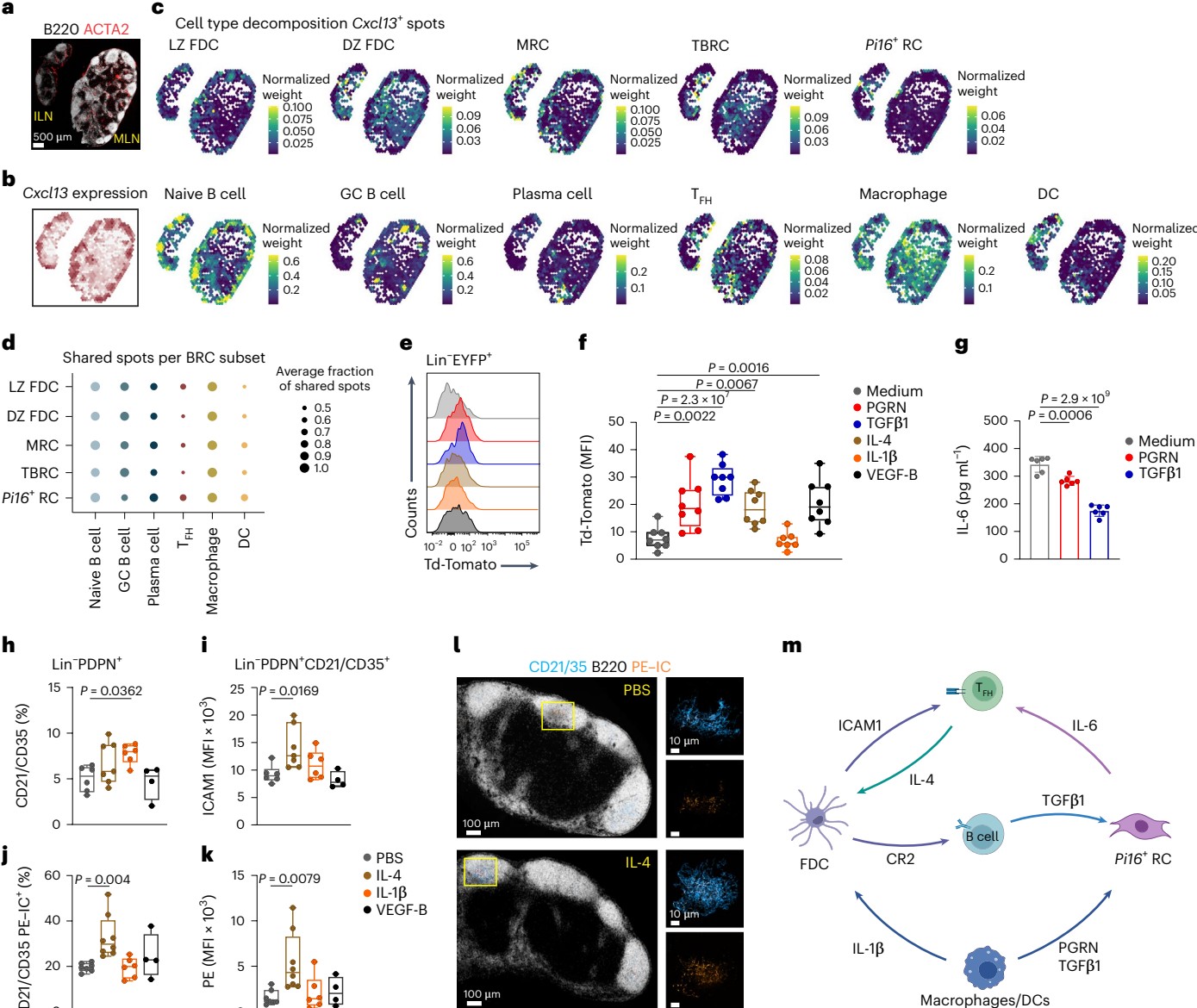

**Fig. 5 | Leukocyte-provided maturation factors specify BRC subset identity.**
**a**, B220 and ACTA2 immunostaining of ILNs and mesenteric LNs (MLNs) processed
for spatial transcriptomics. **b**, Spatial expression of *Cxcl13*. **c**, Normalized
weights from cell type decomposition projected onto *Cxcl13*⁺ spots. **d**, Dot
plot visualizing the fraction of shared spots for each BRC subset with different
immune cells averaged across four samples. **e,f**, Representative histograms (**e**)
and quantification (**f**) of TdTomato expression in Lin⁻Eyfp⁺ cells from Cxcl13-Cre/
TdTomato EYFP LN fibroblast cultures 48 h after stimulation with the indicated
proteins (Lin⁻ refers to CD45⁻CD31⁻). **g**, IL-6 concentration in supernatants from
LN fibroblast cultures 48 h after stimulation with the indicated factors. **h–l**, In vivo
stimulation with predicted maturation cues. **h,i**, Flow cytometry quantification of
FDC frequencies in Lin⁻PDPN⁺ cells (**h**) and ICAM1 expression in FDCs (**i**). **j,k**, Flow
cytometry quantification of the frequency (**j**) and mean fluorescence intensity
(MFI) (**k**) of PE–ICs on CD21/35⁺ cells after in vivo stimulation. **l**, Representative
confocal microscopy images of PE–IC deposition in LNs from PBS or IL-4-treated
mice. **m**, Schematic representation of BRC-provided niche factors and immune
cell-derived BRC activation and differentiation cues. In **a–d** spatial transcriptome

analysis was performed on *n* = 4 LNs, with a technical replicate for each sample.
In **a–c** one representative sample is shown. In **e,f** data from *n* = 8 replicates from
two independent experiments are shown. The boxplot midline demarcates the
median; the box limits demarcate the upper and lower quartiles; and the whiskers
depict the minimum and maximum. In **g** data from *n* = 6 replicates from two
independent experiments with mean and s.d. are shown. In **h,i** data from *n* = 6 PBS-
treated mice, *n* = 7 recombinant IL-4-treated mice, *n* = 6 recombinant IL-1β-treated
mice and *n* = 4 VEGF-B-treated mice are shown (two independent experiments).
In **j,k** data from *n* = 7 PBS-treated mice, *n* = 8 recombinant IL-4-treated mice, *n* = 6
recombinant IL-1β-treated mice and *n* = 4 VEGF-B-treated mice are shown (two
independent experiments). In **h–k** the midline of the boxplot demarcates the
median; the box limits demarcate the upper and lower quartiles; the whiskers
depict the minimum and maximum. In **l** the representatives of three mice per
treatment are shown. In **f–k** adjusted *P* values were derived from a Dunnett's
multiple comparison test with a 95% confidence interval using a one-way analysis
of variance. **m**, Created with BioRender.

paradigm whereby circulating immune cells imprint BF niches in an
organ-indiscriminate manner thereby securing efficient humoral
immunity across SLOs.

Apart from LTβR-mediated and tumor necrosis factor
receptor-mediated BRC differentiation[5,26], knowledge about BRC sub-
set specification is limited. Our study identified IL-4, VEGF-B, PRGN and

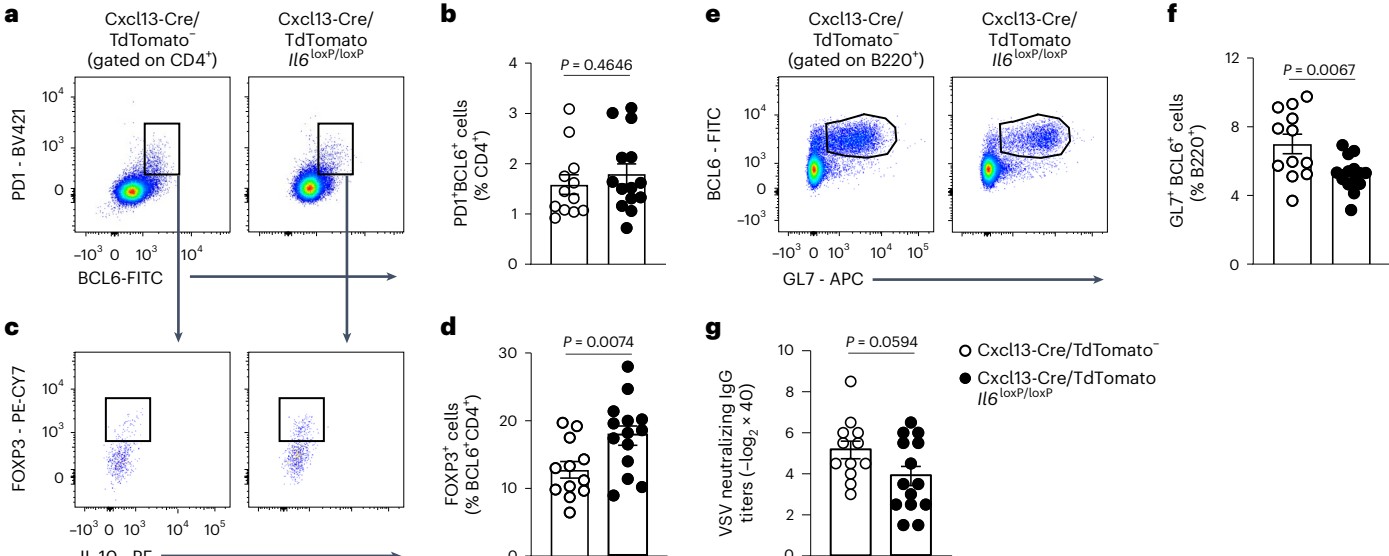

**Fig. 6 | Cxcl13-Cre-provided *Il6* controls T_FH differentiation to sustain GC responses. a,c,e,** Representative flow cytometry plots depicting the gating of T_FH cells (**a**), FOXP3⁺ T_FR cells (**c**) and GC B cells (**e**) according to the indicated markers. **b,d,f,** Quantification of the frequency of T_FH cells (**b**), FOXP3⁺ T_FR cells (**d**) and GC B cells (**f**) in LNs from Cxcl13-Cre/TdTomato⁻ or Cxcl13-Cre/TdTomato *Il6^{loxP/loxP}* mice 14 d after VSV infection. **g,** Quantification of neutralizing serum antibodies

from day 14, VSV-infected Cxcl13-Cre/TdTomato⁻ or Cxcl13-Cre/TdTomato *Il6^{loxP/loxP}* mice. In **a–g** data represent three independent experiments with *n* = 12 Cxcl13-Cre/TdTomato⁻ mice and *n* = 14 Cxcl13-Cre/TdTomato *Il6^{loxP/loxP}* mice. In **b,d,f,g** *P* values were computed using a two-sided, unpaired *t*-test. Data are presented as the mean ± s.d.

TGFβ1 as drivers of CXCL13 production, signifying that redundant cues can maintain B cell niches in lymphoid tissues. Moreover, IL-1β and IL-4 were conserved FDC differentiation and activation cues, respectively. Stromal cell IL-4R expression has been implicated in both cell-intrinsic CXCL12 production[38] and MBC differentiation[39], suggesting that certain molecular circuits may serve dual functions in sustaining FRC differentiation and activation or immune cell function. TGFβ1 and PRGN were identified as factors that impact *PI16*⁺ RC activity in terms of IL-6 production. Interestingly, the regulation of IL-6 production by TGFβ is influenced by the inflammatory milieu and different isoforms of TGFβ. While our study demonstrates that the stimulation of steady-state mouse LN fibroblasts with TGFβ1 results in a dampening of IL-6 production, de Martin et al. showed that TGFβ3 endorses IL-6 produced by *PI16*⁺ RCs from inflamed human tonsillar tissues[40]. In the proximity of the BF, such fine-tuning of IL-6 production by TGFβ and other factors may refine the GC reaction, driving an increase in T_FR cell numbers over the course of the GC response[36]. Collectively, our study unveiled new immune cell-provided factors that steer BRC subset differentiation and activation. While we implicate IL-6 in GC responses, additional cell-targeted genetic studies are warranted to elaborate the immunological sequelae of BRC-abrogated maturation receptors and niche factors.

*PI16*⁺ fibroblasts have recently been suggested to function as a perivascular progenitor population giving rise to activated

fibroblasts in inflamed tissues[19]. However, their function in SLOs is less clear. Of shared BRC populations, we found that *PI16*⁺ RCs expressed the highest level of *Ltbr* transcripts, presumably to sustain elevated receptor expression in the face of higher cell turnover. Spatial transcriptomics demonstrated that in LNs, *Pi16*⁺ RCs localized to the subcapsular sinus (SCS), interfollicular regions and the medulla, all regions transected by lymphatic or blood vessels and leukocyte trafficking. While our study focuses on BRCs, *PI16*⁺ RCs are also captured in CXCL13⁻ FRCs[40]. The study of human tonsillar FRCs identified transcriptional overlap in *PI16*⁺ RCs[40], suggesting that the identified circuits are recapitulated across the broader *PI16*⁺ RC population. Consistent with their proximity to vessels, *PI16*⁺ RCs were predicted to interact with multiple immune cell types and even undergo inflammation-induced changes in abundance and gene expression profiles in chronically inflamed tonsils[40]. These observations suggest that, unlike BRC subsets, which remain transcriptionally stable in inflammatory microenvironments, *PI16*⁺ RCs form an inflammation-adaptable convergence site for immune cells in SLOs.

In summary, canonical BRC niche factors and immune cell-provided BRC specification cues sustain the BRC niches that orchestrate continued antigen sampling and leukocyte interactions within and across SLOs, underscoring a multitiered hierarchy by which BRCs govern efficient humoral immunity.

**Fig. 7 | Cross-species conservation of BRC–immune cell circuits. a,b,** UMAP representation of *CXCL13*-expressing RCs from human LNs (*n* = 4 patients) and tonsils (*n* = 4 patients) colored by SLO (**a**) or subset identity (**b**). **c,** Heatmap showing the average expression of signature genes across BRC subsets from different organs. **d,** Correlation plot visualizing the cross-species similarity of BRC subsets based on the Spearman correlation of the average expression of the 400 most variable homologs. **e,f,** Representative confocal microscopy images of BFs from human LNs (**e**) and tonsillar crypts (**f**) immunostained for the indicated markers. The arrowheads demarcate CXCL13⁺ or CCL19⁺ cells as indicated. **g,** Subset-specific niche factors derived from significantly enriched GO terms determined by enrichment tests on subset-specific genes. **h,** Subset-specific

niche factors projected as average expression of genes derived from enriched pathways onto the UMAP. **i,** Subset-specific signaling pathways with adjusted *P* values as determined by an enrichment test of differentially expressed genes between human BRC subsets. **j,** Heatmap showing the average expression of conserved receptor–ligand pairs derived from the CellPhoneDB analysis. In **g,i** *P* adjustment was performed using the Benjamini–Hochberg procedure. LN scRNA-seq data are representative of *n* = 4 patients and tonsils are representative of *n* = 4 patients and represent 3,450 *CXCL13*-expressing cells from seven independent experiments. In **e,f** images are representative of four human LNs and two human tonsils.

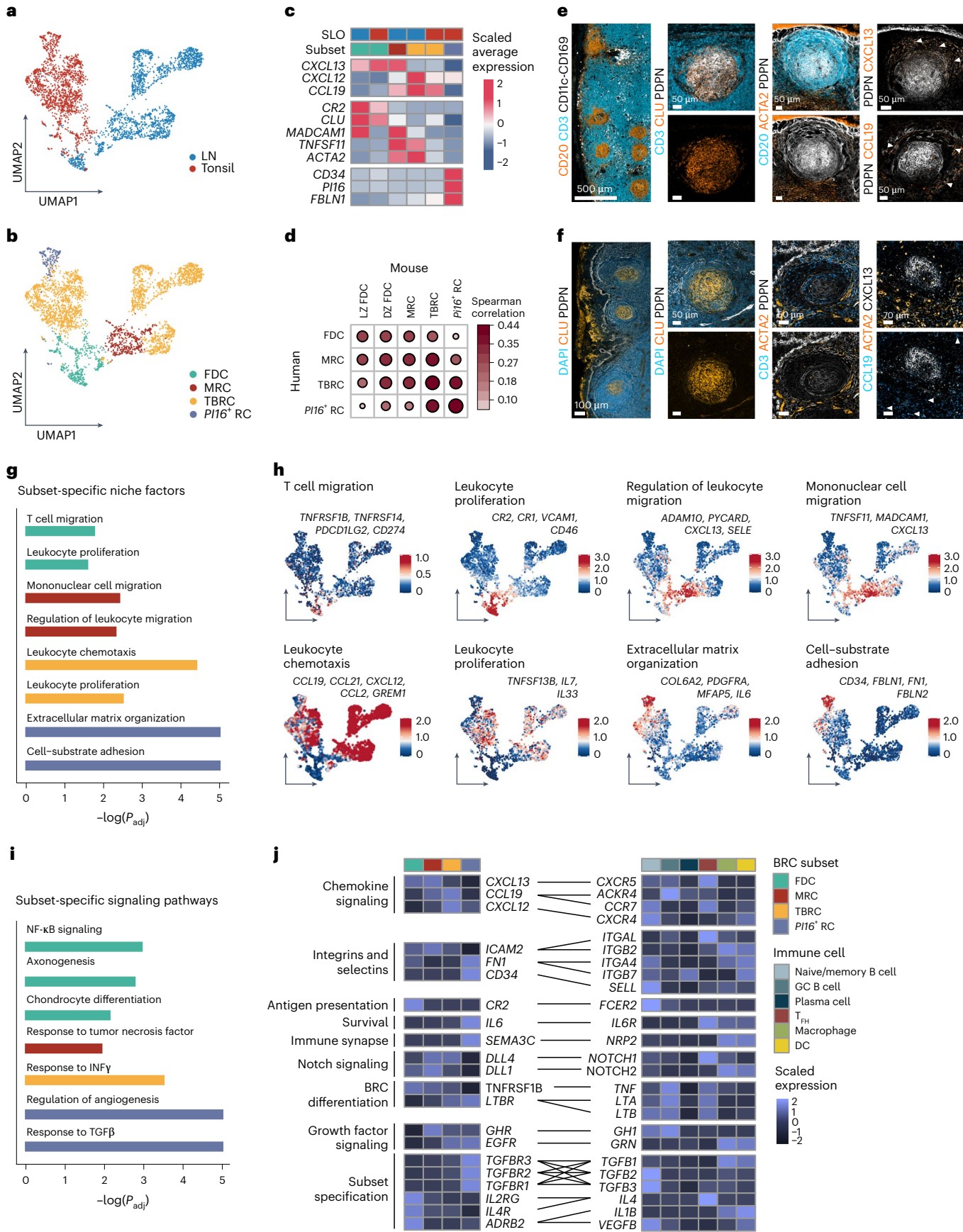

## Online content

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

## Methods

### Mice

The generation of C57BL/6N-Tg(Cxcl13-Cre)723Biat × B6.129×1-Gt (ROSA)26Sortm1(EYFP)-Cos/J (Cxcl13-Cre/TdTomato EYFP) mice was described previously[21]. C57BL/6N-Tg(Cxcl13-Cre)723Biat × B6.129-Il6tm1Jho (Cxcl13-Cre $Il6^{loxP/loxP}$) mice were generated by crossing Cxcl13-Cre mice with B6.129-Il6tm1Jho ($Il6^{loxP/loxP}$) mice[41]. All animals were housed in individually ventilated cages under conventional specific pathogen-free conditions, maintaining a 12 h light–dark cycle, 22 °C ambient temperature and 45–50% humidity. Experiments were performed with 6–10-week-old mice (males and females) in accordance with Swiss federal and cantonal guidelines (Tierschutzgesetz) under permission no. SG/26/2020 granted by the Veterinary Office of the Canton of St. Gallen.

### Human LN and tonsil study participants and sample collection

The study protocol was reviewed and approved by the Ethikkommission Ostschweiz, permission nos. 2019-01532 (LN patients), 2017-00051 (adult tonsil patients) and 2018-01646 (pediatric tonsil patients). All study participants provided written informed consent in accordance with the Declaration of Helsinki (2013) and the International Conference on Harmonization Guidelines for Good Clinical Practice. All regulations were followed according to the Swiss authorities and clinical protocols. Sample size calculation was not performed because of the exploratory nature of the study design.

LN samples were collected from adult patients with benign LN swelling. Tonsil samples were obtained from adult and pediatric patients with obstructive sleep apnea or tonsillitis who underwent routine tonsillectomy (Extended Data Table 1). After surgical excision, lymphoid tissues were collected in sterile PBS on ice and processed within 1 h of resection.

### In vitro differentiation and quantitative PCR of human tonsillar tissue

For in vitro differentiation assays of sorted $PI16^+$ RCs, stromal cells were stained with fluorochrome-conjugated antibodies against human CD45, CD235a, PDPN, CD34, CD31 and UEA1; live CD45⁻CD235a⁻UEA1⁻ CD31⁻PDPN⁺CD34⁺ cells were sorted for in vitro stimulation.

Bulk primary tonsillar fibroblasts and sorted $PI16^+$ RCs were cultured for 10–20 d in Roswell Park Memorial Institute (RPMI) 1640 medium supplemented with 10% FCS (Sigma-Aldrich), 1% penicillin-streptomycin (Sigma-Aldrich) and 16 µg ml⁻¹ gentamicin. To evaluate the differentiation and activation potential, cells were plated at 15,000 cells per cm² and stimulated for 48 h with recombinant human TGFβ1 (1 ng ml⁻¹, catalog no. 7754-BH/CF, R&D Systems), recombinant human PRGN (10 ng ml⁻¹, catalog no. 2420-PG, R&D Systems) or recombinant human GH (10 ng ml⁻¹, catalog no. 1067-GH/CF, R&D Systems). For real-time quantitative PCR (qPCR), total cellular RNA was extracted from cultured and stimulated cells using the Quick-RNA Microprep Kit (catalog no. R1051, Zymo Research) according to the manufacturer's protocol. Complementary DNA (cDNA) was prepared using the High Capacity cDNA Reverse Transcription Kit (catalog no. 4368814, Applied Biosystems) and qPCR was performed using the PowerUp SYBR Green Master Mix (Applied Biosystems) on a QuantStudio 5 system (Applied Biosystems). Expression levels were measured with $PI16$ (no. QT00007000, QIAGEN) and $SOX9$ (no. QT00001498, QIAGEN) primers.

### VSV infection

For scRNA-seq, mice were immunized intravenously with $2 \times 10^7$ plaque-forming units (PFUs) of VSV. Draining ILNs and spleens were collected on day 14 after immunization. To assess GC output, mice were immunized subcutaneously in the flanks on day 0 with $2 \times 10^7$ PFU ultraviolet (UV)-inactivated VSV and again on day 4 with $5 \times 10^7$ PFU UV-inactivated VSV. Serum and draining ILNs were collected on day 14 after immunization.

### VSV-specific antibody detection

Neutralizing VSV-specific antibody titers in sera were measured as described previously[42]. Briefly, IgG titers were determined by incubating sera with 0.1 M 2-mercaptoethanol in PBS for 1 h at room temperature before dilution. Then, 1:2 dilutions of serum from Cxcl13-Cre⁻ or Cxcl13-Cre $Il6^{loxP/loxP}$ immunized mice were mixed with 50 PFU VSV in 96-well flat bottom tissue culture plates seeded with Vero cells and incubated for 24 h at 37 °C. Neutralizing titers were taken as the dilution that resulted in a 50% reduction of viral plaques.

### Preparation of fibroblasts for in vitro cultures and in vitro stimulation

MLNs from Cxcl13-Cre/TdTomato mice were collected and mechanically dissociated in a 24-well dish filled with RPMI 1640 medium containing 2% FCS, 20 mM HEPES (all from Lonza), 1 mg ml⁻¹ collagenase P (Sigma-Aldrich), 25 µg ml⁻¹ DNase I (Applichem) and dispase (Roche). Dissociated tissues were incubated at 37 °C for 30 min. After enzymatic digestion, cell suspensions were washed with PBS containing 0.5% FCS and 10 mM EDTA. For the in vitro assays, LN fibroblasts were cultured for 10 d in RPMI 1640 supplemented with 10% FCS, 1% penicillin-streptomycin and 16 µg ml⁻¹ gentamicin. To evaluate the differentiation and activation potential, $1.5 \times 10^4$ cells were plated in 24-well plates and stimulated for 48 h with recombinant mouse TGFβ1 (10 ng ml⁻¹, catalog no. 7666-MB/CF, R&D Systems), recombinant mouse PRGN (10 ng ml⁻¹, catalog no. 2557-PG, R&D Systems), recombinant mouse VEGF (25 ng ml⁻¹, catalog no. 493-MV/CF, R&D Systems), recombinant mouse IL-4 (10 ng ml⁻¹, catalog no. 404-ML/ CF, R&D Systems) or recombinant mouse IL-1β (1 ng ml⁻¹, catalog no. ab259421, Abcam). Supernatants were collected to determine the IL-6 concentration using the Mouse Inflammation Cytometric Bead Array Kit (BD Biosciences) according to the manufacturer's instructions. Cells were collected and the expression of TdTomato on EYFP⁺PDPN⁺ cells was determined by flow cytometry.

### FDC in vivo activation and immune complex capture

Mice were injected subcutaneously in both flanks with PBS (50 µl), recombinant mouse VEGF (50 ng), recombinant mouse IL-4 (70 ng) or recombinant mouse IL-1β (10 ng) for three consecutive days. After the last subcutaneous injection, mice received PE–ICs. PE–ICs were prepared by mixing 30 µg of R-phycoerythrin (ANASPEC) with 30 µg of mouse-anti-PE (catalog no. PE001, BioLegend); the mix was incubated for 15 min at 37 °C in the dark. PE–ICs were injected subcutaneously in both flanks in a 50-µl volume. Then, 48 h after the last stimulation injection both ILN were collected and single-cell suspensions were generated for flow cytometry analysis as described previously[5].

### Stromal and hematopoietic cell isolation

For stromal cell preparation from murine LNs, spleen and Peyer's patches, tissues were cut into small pieces and collected in RPMI 1640 medium containing 2% FCS, 20 mM HEPES pH 7.2 (Lonza), 0.375 mg ml⁻¹ collagenase P and 25 µg ml⁻¹ DNase I. Dissociated tissue was incubated at 37 °C for 60 min, with resuspension and collection of the supernatant every 15 min. For the cell-sorting experiments, stromal cells were enriched by incubating the cell suspension with MACS anti-CD45 and anti-TER119 microbeads (Miltenyi Biotec) and passing it through a MACS LS column (Miltenyi Biotec). The unbound single-cell suspension was stained for further flow cytometry analysis or cell sorting.

For enzymatic dissociation of human LNs, torn tissues were collected in RPMI 1640 medium containing 2% FCS, 20 mM HEPES pH 7.2, 2.4 mg ml⁻¹ dispase, 0.375 mg ml⁻¹ collagenase P and 25 µg ml⁻¹ DNase I. For tonsils, heavily clotted or cauterized areas of the tissue were removed and tonsils were then dissected into small pieces. To isolate tonsillar stromal cells, tissues were digested with the MACS Human Tumor Dissociation Kit (Miltenyi Biotec) according to the manufacturer's protocol. For both LN and tonsillar tissues, to enrich for

stromal cells, hematopoietic cells and erythrocytes were depleted by incubating the cell suspension with MACS anti-CD45 and anti-CD235a microbeads (Miltenyi Biotec) and passing it through a MACS LS column. The unbound single-cell suspension was stained for further cell sorting.

To isolate hematopoietic cells from murine or human lymphoid organs, tissues were gently smashed across a 26-gauge wire mesh and washed with medium (RMPI 1640 containing 2% FCS, 1% penicillin-streptomycin and 20 mM HEPES) until further staining for flow cytometry analysis or cell sorting.

## Flow cytometry

Cell suspensions were incubated for 20 min at 4 °C in PBS containing 1% FCS and 10 mM EDTA with the following antibodies: anti-mouse CD45, anti-mouse CD31, anti-mouse PDPN, anti-mouse CD21/CD35, anti-mouse SCA1, anti-mouse B220, anti-mouse CD4, anti-mouse CD8, anti-mouse CD19, anti-mouse CD38, anti-mouse GL7 and anti-mouse CD11b (all from BioLegend); anti-mouse CD157 and anti-human CD45 (both from BD Biosciences); anti-human PDPN and anti-human CD31 (both from Thermo Fisher Scientific); and anti-human EPCAM, anti-human CD14, anti-human CD3 and anti-human CD19 (all from BioLegend). LIVE/DEAD cell discrimination was performed either by using a fixable BV510 Dead Cell Staining Kit (Molecular Probes) before antibody staining or by adding 7-aminoactinomycin D (Calbiochem) before acquisition. Cells were acquired with an LSR Fortessa (BD Biosciences) and analyzed with the FlowJo (v.10) software (FlowJo LLC) according to established guidelines. Cell sorting was performed using a BD FACSMelody Cell Sorter and the FACSChorus (v.1.3) software (BD Biosciences).

## Immunofluorescence confocal microscopy

Tissues were processed for either vibratome or cryotome sectioning. For vibratome processing, tissues were fixed for 4 h at 19 °C in freshly prepared 4% paraformaldehyde (Merck Millipore) under agitation. Organs were embedded in 4% low-melting agarose (Invitrogen) in PBS and serially sectioned with a vibratome (VT-1200, Leica Biosystems). Then, 40-μm thick sections were blocked in PBS containing 10% FCS, 1 mg ml$^{-1}$ anti-Fcγ receptor (BD Biosciences) and 0.1% Triton X-100 (Sigma-Aldrich). Human tonsils and murine SLOs stained with CXCL13$^+$FNDC1$^+$ were fresh-frozen and sectioned by cryotome. Tissues were immediately embedded in FSC 22 Clear (Leica Biosystems) and frozen in an isopropanol-dry ice bath and stored at −80 °C. Then, 10-μm sections were mounted onto slides (Thermo Fisher Scientific) and fixed for 1 min in methanol at −20 °C. Tissues were blocked in PBS containing 10% FCS, 1 mg ml$^{-1}$ anti-Fcγ receptor and 0.1% Triton X-100, and stained overnight at 4 °C with the indicated antibodies. Unconjugated and biotinylated antibodies were stained with the indicated secondary antibodies or streptavidin conjugates. Microscopy was performed using a confocal microscope (LSM-980, ZEISS) and images were recorded and processed with the ZEN blue software (v.3.3, ZEISS). Imaris v.9 was used for image analysis.

## Droplet-based scRNA-seq

Cells isolated from murine LNs, splenic WP or Peyer's patches were sorted for CD45$^-$TdTomato$^+$EYFP$^+$ cells or CD45$^+$ cell populations. Isolated hematopoietic, as well as non-hematopoietic, non-endothelial cells from human LNs or tonsillar tissue were sorted as indicated (Extended Data Figs. 7 and 8 and Extended Data Table 1). Sorted single-cell suspensions were emulsified for library generation using the droplet-based 10X Chromium (10X Genomics) system[43]. The cDNA libraries were generated according to the established commercial protocol for Chromium Single Cell 3′ Reagent Kit (v.3 Chemistry) and sequenced by the NovaSeq 6000 system (Illumina) at the Functional Genomic Center Zurich. To get sufficient numbers of cells across organs and conditions, for murine BRCs a total of 19 samples (LN: seven samples; spleen: seven samples; Peyer's patch: five samples) were processed

in eight batches with batches spanning multiple organs. For murine immune cells, six samples from immunized mice were processed (LN: two samples; spleen: two samples; Peyer's patch: two samples) in one batch. Human samples were collected as indicated in the patient summary table (Extended Data Table 1). All samples from the same patient were processed in the same batch.

Gene expression estimation from sequencing files was done using CellRanger (v.3.0.2)[44] count with the Ensembl GRCm38.9 release as reference to build the index for murine samples, and GRCh38.9 used as reference for the human samples. Quality control was performed in R v.4.0.0 using the R/Bioconductor package scater (v.1.16.0)[45] and included removal of damaged and contaminating cells based on (1) very high or low unique molecular identifier (UMI) counts (>2.5 median absolute deviation from the median across all cells), (2) very high or low total number of detected genes (>2.5 median absolute deviation from the median across all cells) and (3) high mitochondrial gene content (>2.5 median absolute deviations above the median across all cells). In the murine BRC samples, only *Cxcl13*-expressing cells were kept for downstream analysis whereas cells expressing one of the markers *Ptprc*, *Cd79a*, *Cd3e*, *Pecam1*, *Lyve1* or *Cldn5* were removed as contaminants. In the human BRC samples, only cells expressing *CXCL13*, but not *CD3E*, *MKI67*, *PTPRC*, *CD79A*, *LYVE1*, *PECAM1* or *MYH11* were kept for downstream analysis. After quality control, the final dataset included 34,538 murine BRCs (naive LN: 14,731, immunized LN: 5,030; naive spleen: 6,996; immunized spleen: 3,365; Peyer's patch: 4,416), 30,766 murine immune cells (LN: 12,351; spleen: 10,554; Peyer's patch: 7,861), 3,450 human BRCs (LN: 1,891; tonsil:1,559) and 135,625 human immune cells (LN: 84,558; tonsil: 51,067).

## Comparative analysis and interactome analysis of murine data

Downstream analysis of BRCs was performed using Seurat (v.4.0.1)[46,47] and included normalization, scaling, dimensionality reduction with principal component analysis (PCA) and UMAP, graph-based clustering and calculation of unbiased cluster markers. Clusters were characterized based on the expression of calculated cluster markers and canonical marker genes as reported in previous publications[5,24,26]. After cluster characterization for each organ individually, BRC samples from all organs were merged and integrated across organs to compare subset identities independent of their organ identity and to confirm the presence of shared BRC subsets. Integration was performed using the IntegrateData function from the Seurat package.

Further comparative analysis was run on shared BRC subsets only and included determination of organ-specific and subset-specific gene signatures, where organ-specific and subset-specific genes were calculated running the FindConservedMarkers function from Seurat. Functional signatures were further derived by running GO enrichment analysis on organ-specific or subset-specific genes using the cluster-Profiler R/Bioconductor package (v.3.15.3)[48]. Differences between immunized and naive BRCs were determined as differentially expressed genes for BRCs from LNs and splenic WP individually. Factors driving gene expression were examined by running variance partition analysis using the fitExtractVarPartModel function from the variancePartition R package (v.1.22.0)[49] and MDS on pseudobulks derived from the aggregateData function of the muscat R/Bioconductor package (v.1.6.0)[50], as well as dimensionality reduction with diffusionmap as implemented in the scater R/Bioconductor package (v.1.16.0)[45].

Before analyzing BRC−immune cell crosstalk, immune cells from all organs were merged and characterized including normalization, scaling, dimensionality reduction with PCA and UMAP, graph-based clustering and calculation of unbiased cluster markers. After cluster characterization, only immune cells known to be situated in and around the BF and BRCs from immunized mice were kept for interactome analysis. BRC−immune cell interactions were then predicted using CellPhoneDB (v.2.1.7)[51] as a tool for interactome analysis based on gene

expression data. Immune cells and BRCs were first downsampled to no more than 500 cells per cell type or subset before CellPhoneDB was run with Python v.3.7.0 and default parameters. For significant interactions, downstream signaling patterns were examined and projected based on the average expression of all genes comprising individual signaling pathways derived from the BioCarta[52] or WikiPathways[53] signaling pathway collection.

### Comparative analysis and interactome analysis of human data
Human BRCs were first analyzed for each organ individually before they were merged and compared across organs. Analysis was performed with Seurat (v.4.0.1)[46,47] and included normalization, scaling, dimensionality reduction with PCA and UMAP, graph-based clustering and cluster characterization. Cross-species similarity of BRC subsets was analyzed by combining the murine and human data based on homologs and calculating the Spearman corelation of the average expression of the 400 most variable genes.

Comparative analysis was performed as for the murine data, including determination of organ-specific and subset-specific gene signatures, variance partition analysis and MDS. Immune cells were merged, characterized and filtered for immune cell types known to be involved in the initiation of humoral immunity in SLOs. Finally, BRC–immune cell interactions were predicted using CellPhoneDB (v.2.1.7)[51] and downstream signaling patterns were examined based on the BioCarta[52] and WikiPathways[53] signaling pathway collections.

### Sample preparation and processing for Visium spatial transcriptomics
The processing of spatial transcriptomics followed the commercial brochure of the Visium Gene Expression Kits (catalog no. PN-1000184, 10X Genomics). Briefly, freshly isolated ILNs and MLNs were frozen in 2-methylbutane (catalog no. M32631, Sigma-Aldrich) with liquid nitrogen and immediately embedded into frozen section media (FSC22, catalog no. 3801480, Leica Biosystems) for spatial transcriptomics. The tissue blocks were cut into 10-µm sections using a cryostat and mounted on Visium Gene Expression slides (catalog no. PN-1000188, 10X Genomics). Sections on Visium Gene Expression slides were first fixed with precooled methanol for 30 min and stained with 4′,6-diamidino-2-phenylindole, B220 and ACTA2 and imaged with confocal microscopy to visualize the fiducial frames of capture areas and cell compartmentalization in different sections. The optimal permeabilization time for in situ hybridization and reverse transcription of murine LN transcripts was 15 min evaluated with the 10X Genomics Visium Tissue Optimization Kit (catalog no. PN-1000193, 10X Genomics). cDNA libraries were generated using the Visium Spatial Gene Expression 3′ Library Construction Kit (catalog no. PN-1000190, 10X Genomics) and were sequenced on an Illumina NovaSeq SP flowcell by the Functional Genomic Center in Zurich.

### Visium spatial transcriptomic analysis
Fiducial frames around the capture area on the Visium slide were aligned manually using the Loupe Browser (v.6.2.0) and the high-resolution images of the tissue. Next, gene expression estimation from sequencing files was done using the Space Ranger software (v.1.3.1) with the Ensembl GRCm38.9 release as the reference genome and the manually aligned images as loupe-alignment input. Further analysis was performed in R v.4.2.1 using the R/Bioconductor package SpatialExperiment (v.1.6.1)[54]. Quality control included removal of spots with (1) very high or low UMI counts (>3 median absolute deviation from the median across all spots), (2) very high or low total number of detected genes (>3 median absolute deviation from the median across all spots) and (3) high mitochondrial gene content (>10 median absolute deviations above the median across all spots). Log-transformed normalized counts were calculated and only *Cxcl13*-expressing spots were included in the downstream analysis.

### Cell type decomposition of spatial transcriptomes
After preprocessing and quality control, the R package spacexr (v.2.0.1)[55] was used to infer the cell type compositions of Visium spots based on an scRNA-seq reference dataset. The reference was built by merging BRC and immune cell datasets from the murine LNs used in this study. Based on this reference, cell type decomposition was performed by running the run.RCTD function from the spacexr package using doublet_mode = 'full' to derive normalized weights for all cell types in the reference. Cell type neighborhoods were explored by quantifying the fraction of shared spots between all BRC subsets and immune cells, where a cell type was considered to be present in a spot if the computed normalized weight was greater than 0.01.

### Statistics and reproducibility
For the stimulation experiments, animals were randomly assigned to groups before the start of experimentation. The investigators were blinded to allocation during the experiments and outcome assessment. No data were excluded from the analyses. Statistical assumptions, including data distribution, independence of observations and homogeneity of variance, were considered for each dataset and statistical tests were performed accordingly. Prism 8 (GraphPad Software) was used for statistical analyses. Differences with $P < 0.05$ were considered statistically significant; multiple testing correction was performed as relevant.

### Reporting summary
Further information on research design is available in the Nature Portfolio Reporting Summary linked to this article.

### Data availability
Ensembl GRCm38.9 and GRCh38.9 were used as reference genomes to build the indexes. The scRNA-seq and spatial transcriptomic data generated in this study have been deposited in the BioStudies database (www.ebi.ac.uk/biostudies/). The mouse scRNAseq data are available under accession no. E-MTAB-11738; the human LN data are available under accession no. E-MTAB-11710 and the data from human palatine tonsils are available under accession no. E-MTAB-11715. Spatial transcriptomic data are available under accession no. E-MTAB-12468. The processed data files can be downloaded from the figshare platform at https://doi.org/10.6084/m9.figshare.21221291 and explored via an interactive browser at https://immbiosg.github.io/FRCdataExplorer/.

### Code availability
The code used for data analysis in this project is available at GitHub (https://github.com/mluetge/CrossSLO_BRC_CXCL13).

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

## Acknowledgements

We thank L. Büchler for excellent technical assistance and S. Grabherr for excellent assistance with mouse husbandry. This study received financial support from the Swiss National Science Foundation (grant nos. 177208 and 182583 to B.L. and grant no. 180011 to N.B.P.). N.B.P. was supported by a Hans Peter Hofschneider Professorship awarded by the Stiftung für Experimentelle Biomedizin. Y.S was supported by the Research Commission of the Kantonsspital St.Gallen (grant 19/07). The funders had no role in study design, data collection and analysis, decision to publish or preparation of the manuscript.

## Author contributions

N.B.P. designed the study, performed the experiments, discussed the data and wrote the paper. B.L. designed the study, discussed the data and wrote the paper. M.L. analyzed the data and wrote the paper. A.D.M., C.G.-C., C.P.-S., Y.S., L.O., H.-W.C., L.K. and N.C. performed the experiments and discussed the data. C.S. and M.D.R. discussed the data. S.J.S. discussed the data and provided the material from patients.

## Competing interests

C.P.-S., C.G.-C., H.-W.C., L.O., N.B.P. and B.L. are founders and shareholders of Stromal Therapeutics. L.O. and B.L. are members of the board of Stromal Therapeutics. The other authors declare no competing interests.

## Additional information

**Extended data** is available for this paper at https://doi.org/10.1038/s41590-023-01503-3.

**Correspondence and requests for materials** should be addressed to Burkhard Ludewig or Natalia B. Pikor.

**Peer review information** *Nature Immunology* thanks Kim Good-Jacobson and the other, anonymous, reviewer(s) for their contribution to the peer review of this work. L. A. Dempsey was the primary editor on this article and managed its editorial process and peer review in collaboration with the rest of the editorial team at *Nature Immunology*. Peer reviewer reports are available.

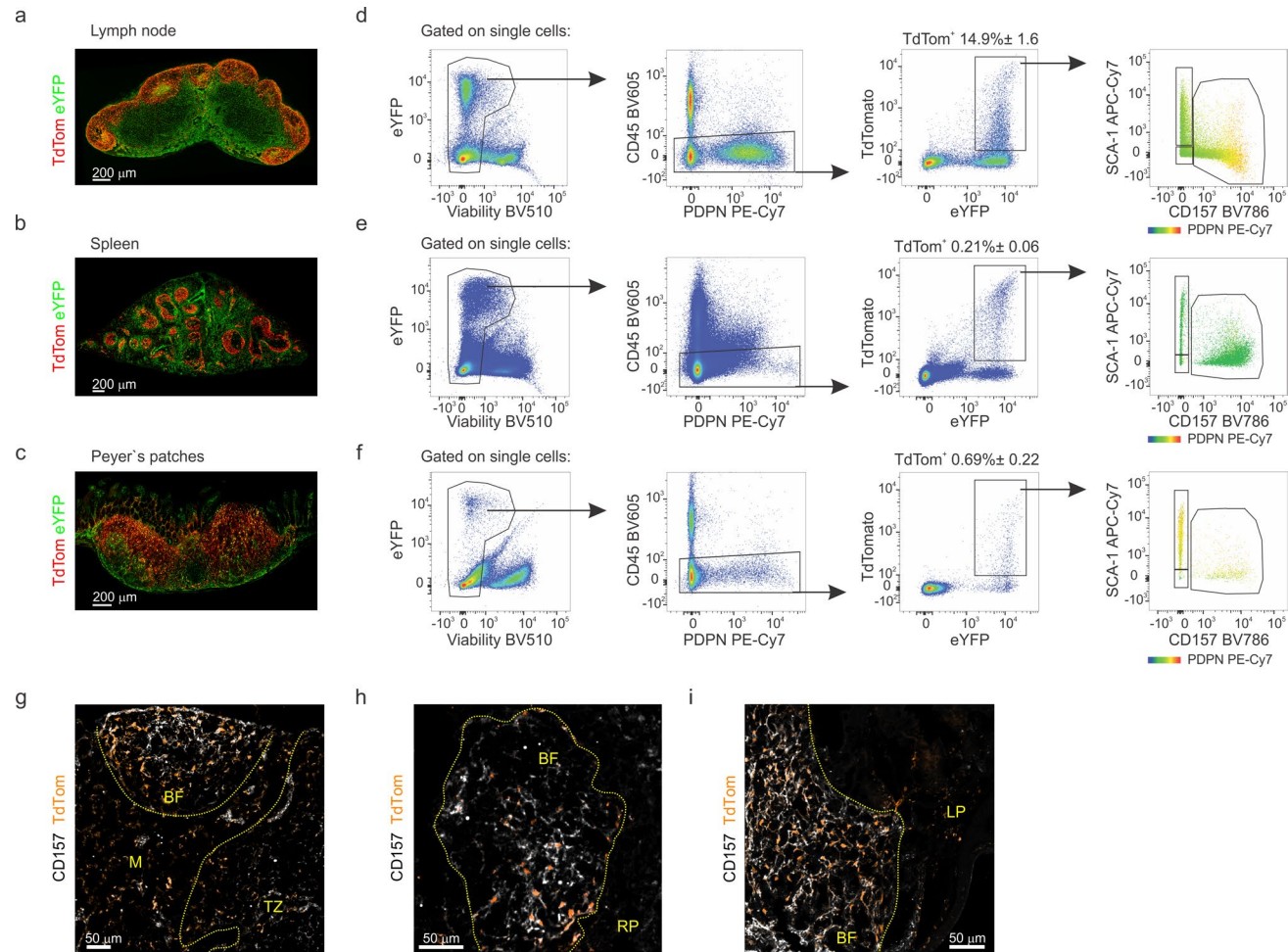

**Extended Data Fig. 1 | Cxcl13-Cre⁺ BRCs in murine SLOs. a–c**, Representative immunofluorescence images of inguinal lymph nodes (**a**), the spleen (**b**), and Peyer's patches (**c**) from naïve Cxc13-Cre/TdTomato EYFP mice. Immunostaining for EYFP and TdTomato identifies Cxcl13-Cre/TdTomato transgene expression. **d-f**, Representative flow cytometric plots depicting the BRC gating strategy for each the inguinal lymph node (**d**), spleen (**e**), and Peyer's patches (**f**) according to the indicated markers. **g-i**, Representative immunofluorescence images of Cxcl13-Cre/TdTomato cells expressing CD157 in the B cell follicle (BF) of the inguinal lymph node (**g**), splenic white pulp (**h**) and Peyer's Patches (**f**). The medulla (M), T cell zone (TZ), red pulp (RP) and lamina propria (LP) are demarcated. In **a-c** data is representative of at least 6 mice. Flow cytometric analysis in **d-f** is representative of n = 6 mice from 2 independent experiments. In **g-i** data is representative of at least 3 mice.

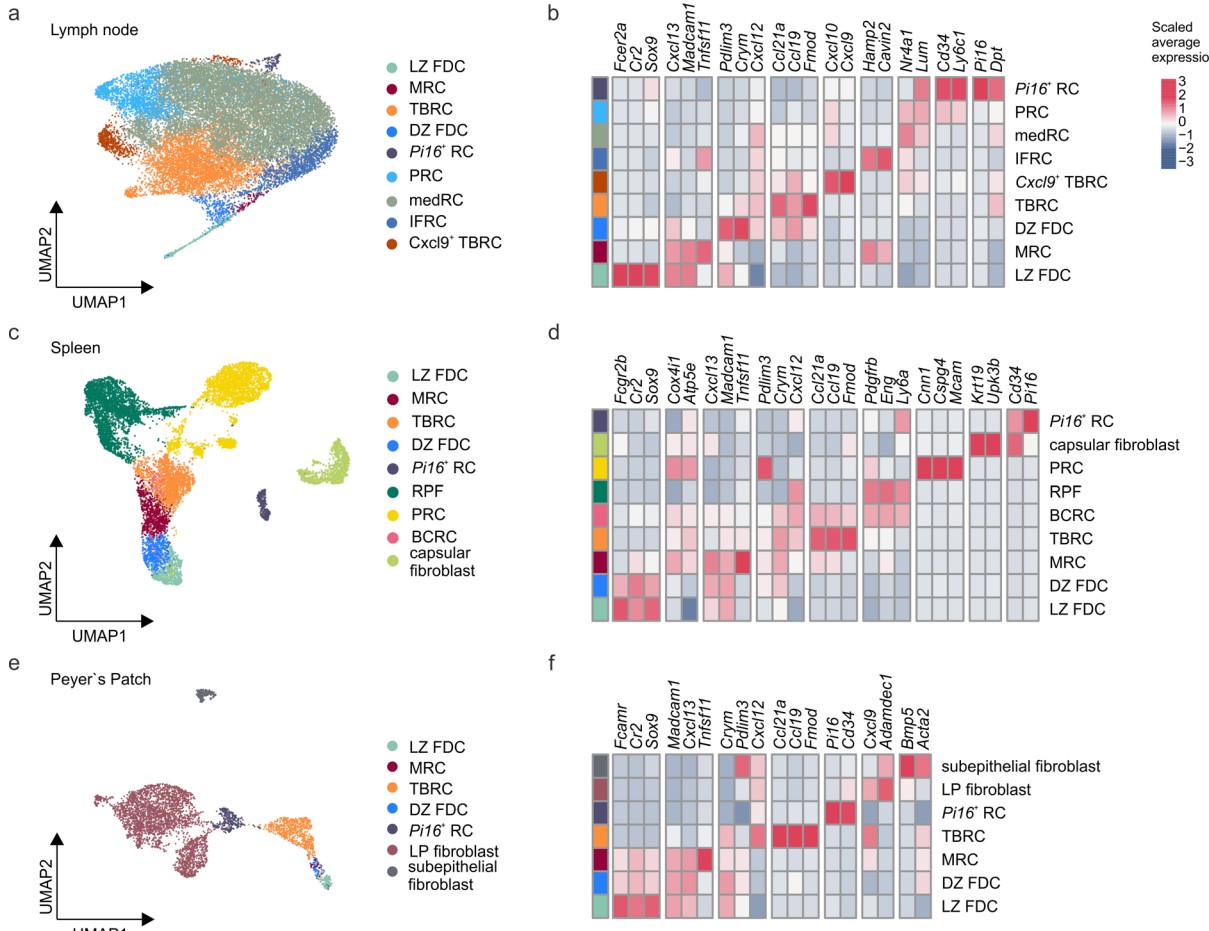

**Extended Data Fig. 2 | Transcriptomic characterization of *Cxcl13*⁺ BRCs in individual SLOs. a**, **c**, **e** UMAP visualizing *Cxcl13*⁺ reticular cells from inguinal lymph nodes (**a**), splenic white pulp (**c**) and Peyer's Patches (**e**) of naïve and VSV-immunized Cxc13-Cre/TdTomato EYFP mice colored by assigned subset identity. **b**, **d**, **f** Heatmaps showing the average expression of signature genes used for characterization of BRC subsets in lymph nodes (**b**), splenic white pulp (**d**) and Peyer's Patches (**f**). In **a**, **b**, scRNA-seq data represent 19,761 *Cxcl13*⁺ cells for n = 5 biological replicates for naive BRCs and 2 biological replicates for immunized mice from 5 independent experiments. In **c**, **d**, scRNA-seq data represent 10,361 *Cxcl13*⁺ cells for n = 4 biological replicates for naive BRCs and 3 biological replicates for immunized mice from 5 independent experiments. In **e**, **f**, scRNA-seq data represent 4,416 *Cxcl13*⁺ cells for n = 5 biological replicates from 5 independent experiments.

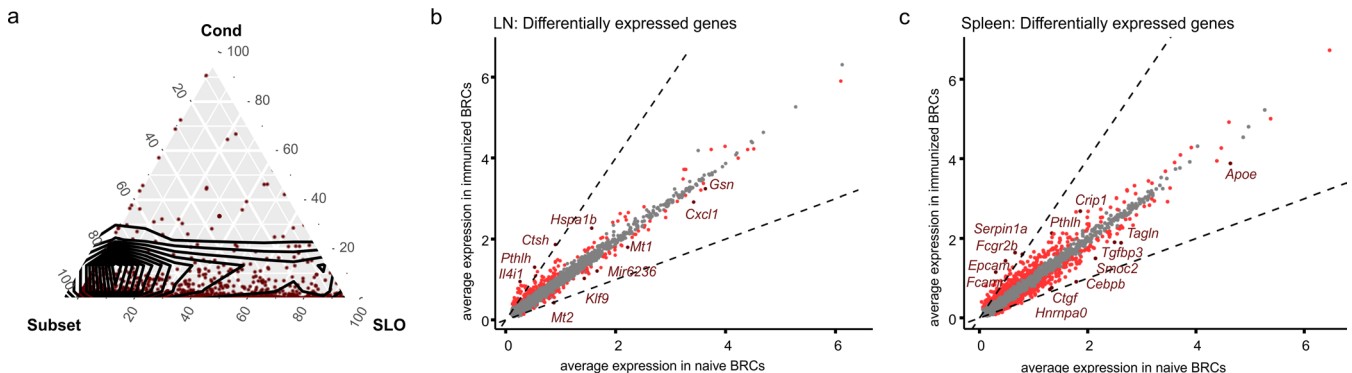

**Extended Data Fig. 3 | Contribution of inflammatory condition on BRC subset identity. a**, Ternary plot visualizing the percentage of variance in the average expression of the 2000 most variable genes explained by either subset identity, SLO or infection condition (Cond). **b**, **c**, Scatterplot comparing the average expression of genes expressed in at least 10 percent of cells within shared BRC subsets in lymph nodes (**b**) and splenic white pulps (**c**) between naïve and immunized mice. Red dots indicate differentially expressed genes with an adjusted p-value < 0.01 and an effect size (logFC) > 0.25. Dotted lines indicate a logFC of 2. In **a**, scRNA-seq data represents 9,572 *Cxcl13*⁺ cells from 9 independent experiments with n = 7 biological replicates for lymph nodes, n = 7 biological replicates for splenic white pulps and n = 5 biological replicates for Peyer's Patches. **b**, scRNA-seq data represents 4,788 *Cxcl13*⁺ cells for n = 5 biological replicates for naive BRCs and 2 biological replicates for immunized mice from 5 independent experiments. In **c**, scRNA-seq data represents 3,632 *Cxcl13*⁺ cells for n = 4 biological replicates for naive BRCs and 3 biological replicates for immunized mice from 5 independent experiments. **b**, **c**, p-value adjustment was performed using Bonferoni correction.

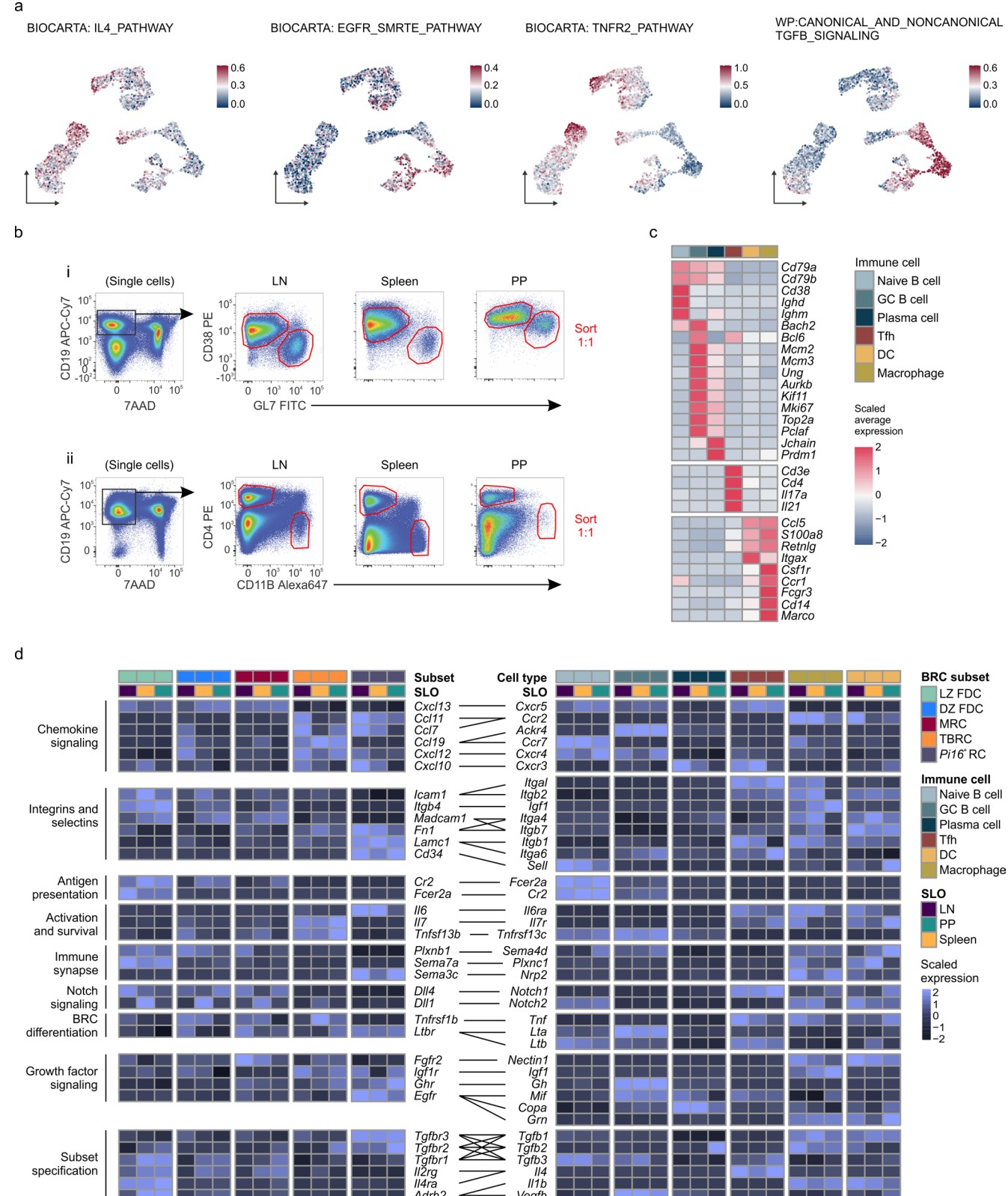

**Extended Data Fig. 4 | See next page for caption.**

**Extended Data Fig. 4 | Conserved expression of BRC-derived niche factors across SLOs. a**, Expression pattern of signaling pathways downstream of subset-specific BRC maturation cues projected as average expression across all genes from the indicated pathways onto the UMAP of shared BRC subsets. **b**, Representative flow cytometric plots depicting the sorting strategy of CD38+ GL7− naïve and CD38−GL7+ GC B cells (i) and CD4+ and CD11b+ myeloid cells (ii) from each the inguinal lymph node, spleen, and Peyer's patches according to the indicated markers. **c**, Heatmap showing the average expression of signature genes used for characterization of immune cells isolated from lymph nodes, spleen, and Peyer's patches. **d**, Heatmap visualizing the average expression of receptor-ligand pairs identified as significant interactions common in all SLOs by CellPhoneDB analysis. In **a**, scRNA-seq data represent 9,572 *Cxcl13*+ cells (downsampled to 3,361 *Cxcl13*+ cells for visualization) from 8 independent experiments with n = 7 biological replicates for lymph nodes, n = 7 biological replicates for splenic white pulps and n = 5 biological replicates for Peyer's Patches. In **b**, **c**, scRNA-seq data represents 23,250 immune cells from 1 independent experiment with n = 4 mice and 9,572 *Cxcl13*+ cells from 8 independent experiments with n = 7 biological replicates for lymph nodes, n = 7 biological replicates for splenic white pulps and n = 5 biological replicates for Peyer's Patches.

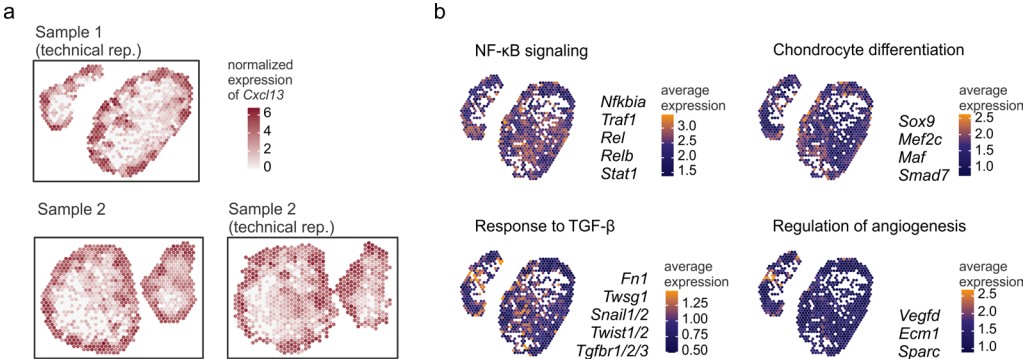

**Extended Data Fig. 5 | Spatial expression of subset-specific BRC intrinsic signaling programs. a**, Spatial expression of *Cxcl13* in two samples and technical replicates showing an inguinal and a mesenteric lymph node each. Sample 1 is shown as representative in Fig. 5b. **b**, Spatial expression of BRC intrinsic signaling signatures derived from conserved subset-specific gene expression profiles and projected on all *Cxcl13*⁺ spots within inguinal and mesenteric lymph nodes from Sample 1 shown in Fig. 5b, c.

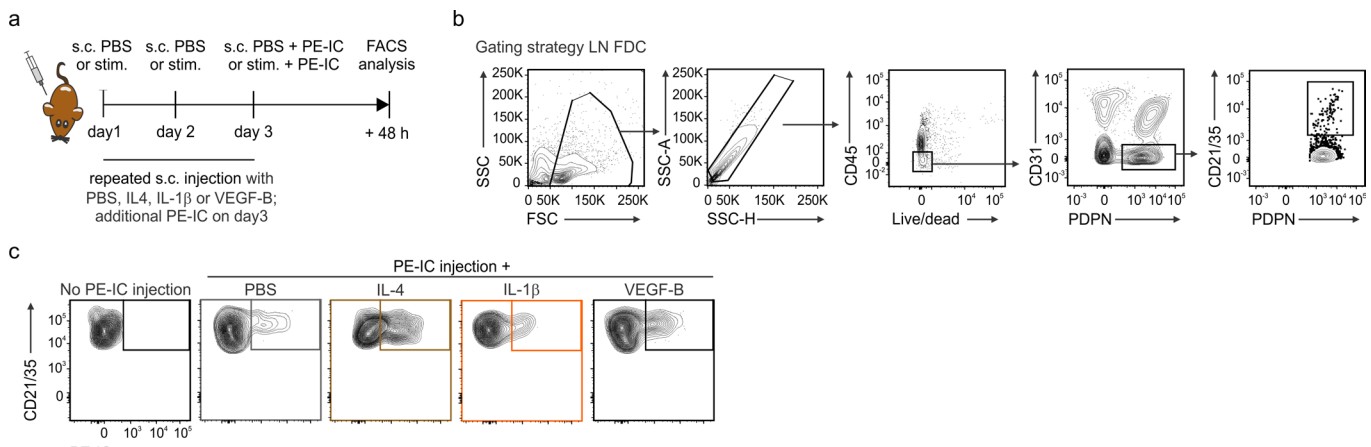

**Extended Data Fig. 6 | Predicted FDC maturation cues impact immune complex binding. a**, Schematic of the experimental workflow for in vivo stimulations with predicted FDC maturation cues. Mice were s.c. injected in both flanks on three consecutive days and FDCs from both inguinal lymph nodes were analysed by flow cytometry two days after the third stimulation. **b**, Representative flow cytometric plots depicting the sorting strategy of CD45⁻CD31⁻PDPN⁺CD21/35⁺ FDCs from inguinal lymph nodes harvested following in vivo stimulations. **c**, Representative flow cytometry plots show the gating used for quantification of PE-IC binding by CD21/35⁺ cells following in vivo stimulations with FDC maturation cues.

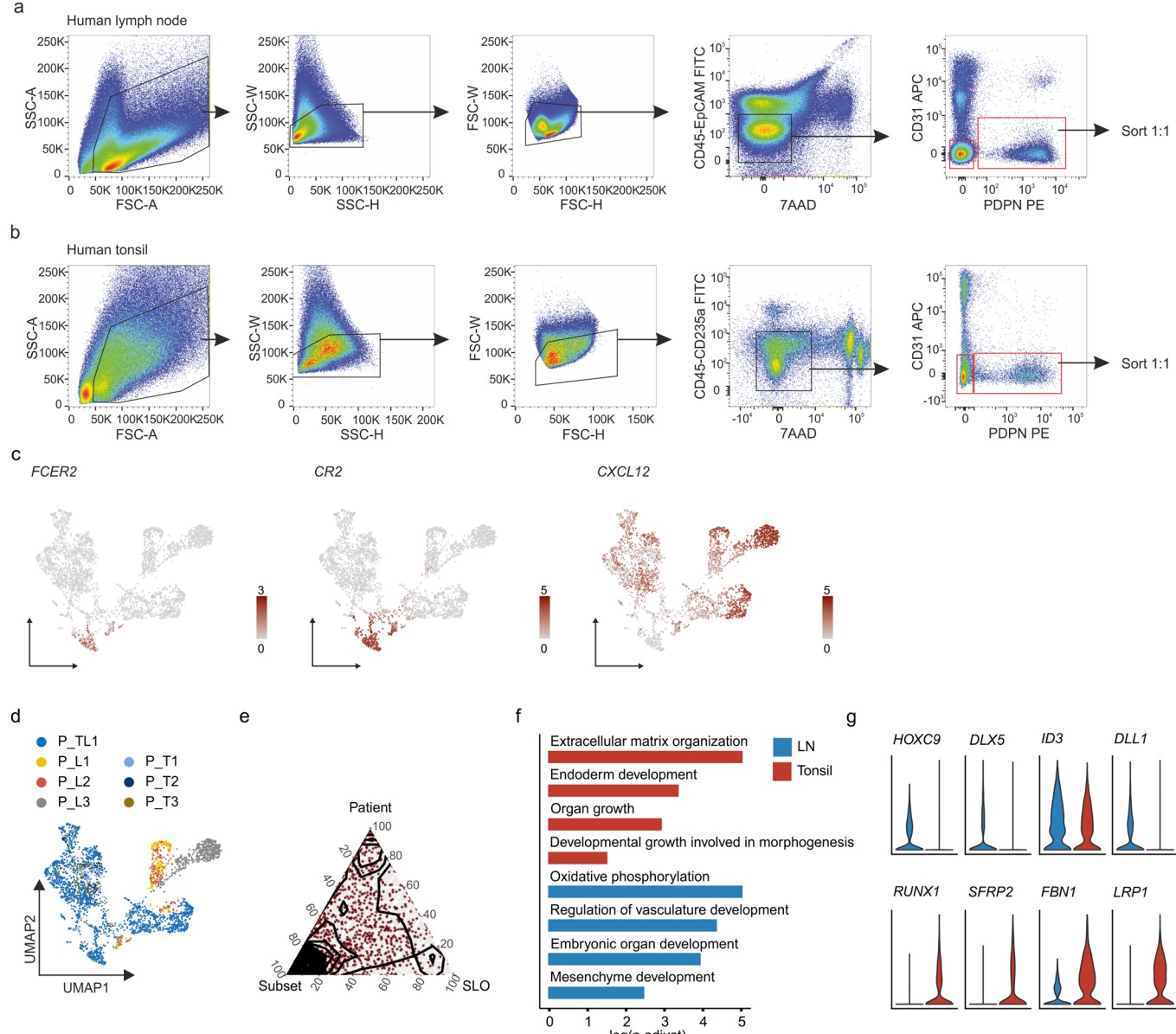

**Extended Data Fig. 7 | Developmental gene signatures imprint human BRC organ-specificity. a**, **b**, Representative flow cytometric plots depicting the sorting strategy of non-hematopoeitic, non-endothelial cells from human lymph nodes (**a**) and human palatine tonsils (**b**) according to the indicated markers. **c**, Feature plots visualizing the expression of the indicated marker genes projected on the UMAP of tonsillar and lymph node BRCs. **d**, UMAP visualizing *CXCL13*⁺ reticular cells from human lymph nodes and palatine tonsils colored by patient identity. **e**, Ternary plot visualizing the percentage of variance in the average expression of the 2000 most variable genes explained by either subset identity, SLO or patient identity. **f**, Organ-specific GO terms with adjusted p-values as determined by enrichment test on differentially expressed genes between SLOs. **g**, Violin plots showing the expression of transcription factors differentially expressed between tonsillar and lymph node BRCs. In **c-g**, data was sampled from n = 4 patients for human lymph nodes and n = 4 patients for human palatine tonsils and represents 3,450 *CXCL13*⁺ cells. **f**, p-value adjustment was performed using Benjamini & Hochberg procedure.

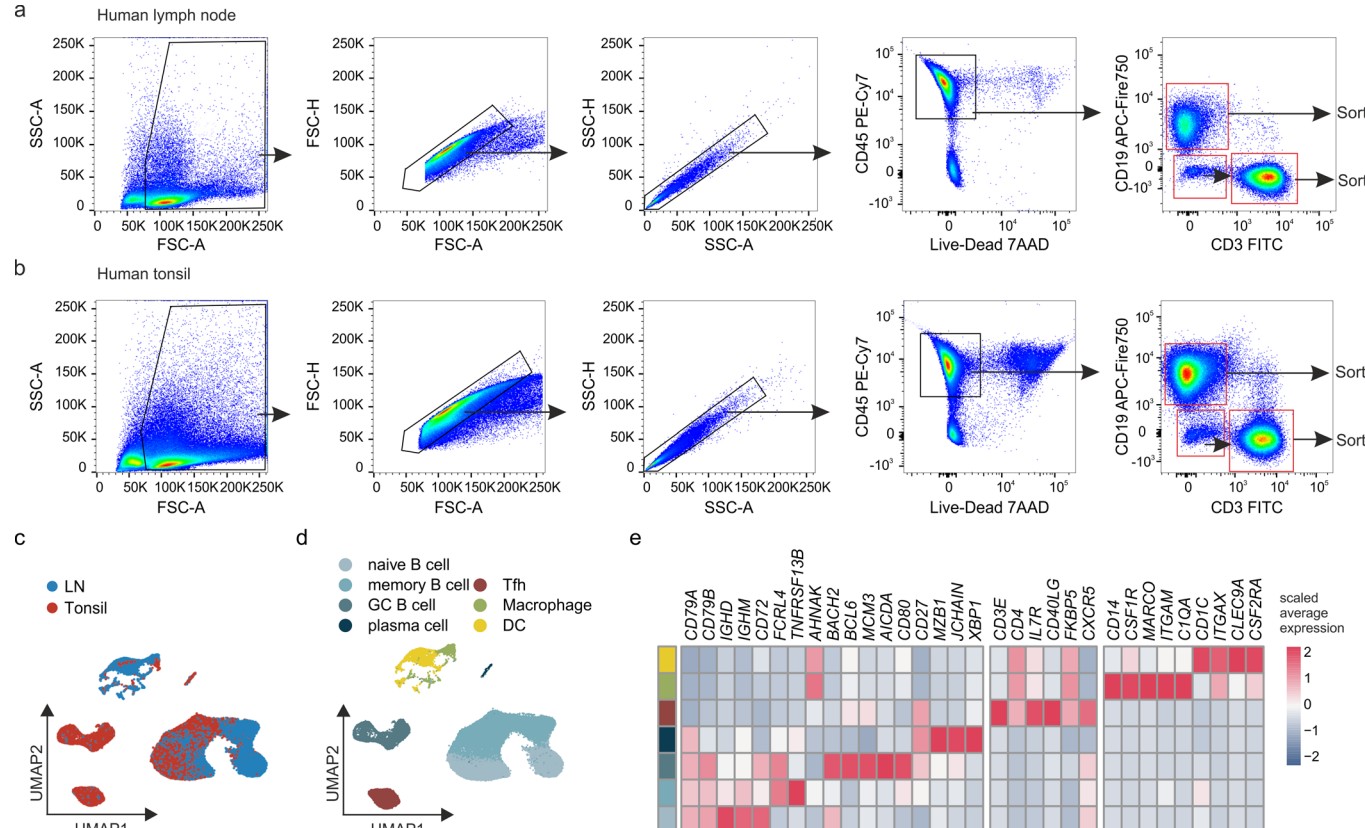

**Extended Data Fig. 8 | Immune cell characterization in human lymphoid tissues. a**, **b**, Representative flow cytometric plots depicting the sorting strategy of CD3⁺ T cells, CD19⁺ B cells, and CD19⁻ CD3⁻ cells from human lymph nodes (**a**) and human palatine tonsils (**b**) according to the indicated markers. **c**, **d**, UMAPs visualizing immune cells known to be situated in and around the B cell follicle from human lymph nodes and human palatine tonsils colored by SLO (**c**) or assigned subset identity (**d**). **e**, Heatmap showing the average expression of signature genes characterizing human immune cells. In **c-e**, data represents 56,887 immune cells sampled from n = 2 patients for human lymph nodes and n = 2 patients for human palatine tonsils.

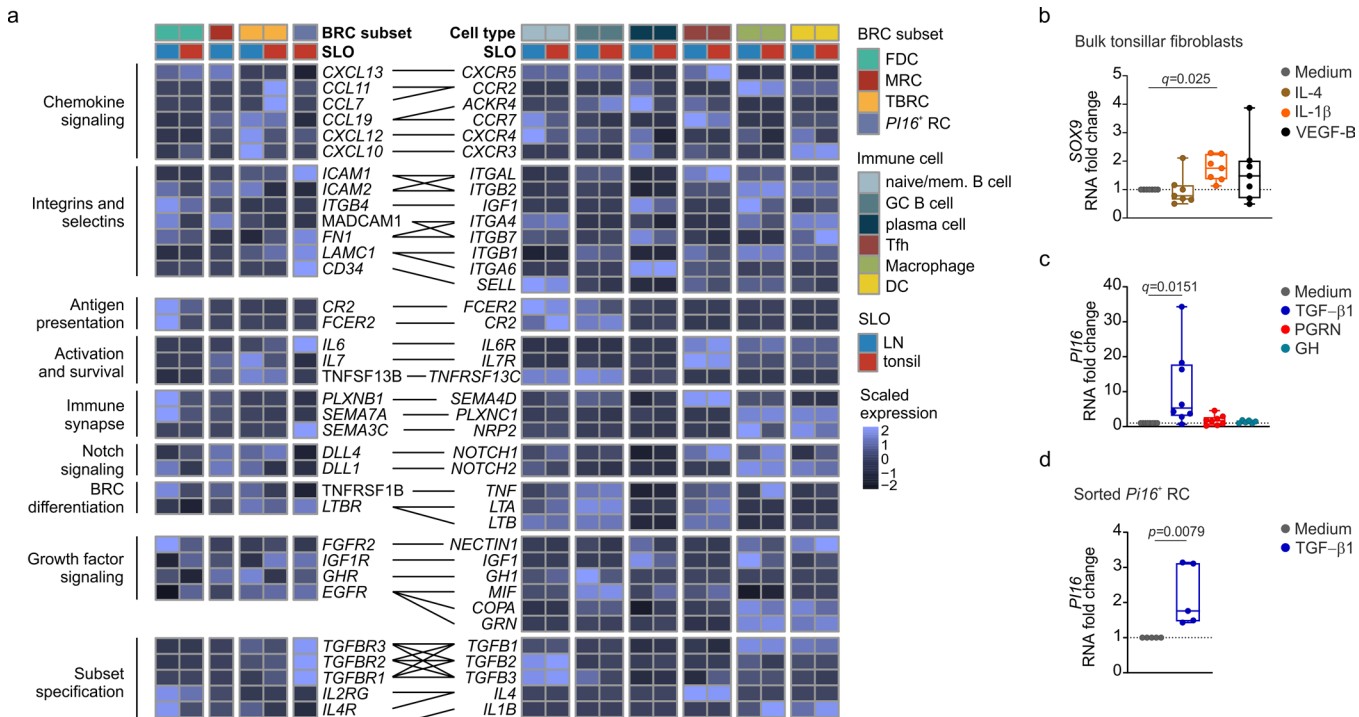

**Extended Data Fig. 9 | BRC-immune cell interaction circuits in human lymphoid tissues. a**, Heatmap visualizing the average expression of conserved receptor-ligand pairs identified as significant interactions in human SLOs by CellPhoneDB analysis. **b**, **c**, Fold changes of *SOX9* (**b**) and *PI16* (**c**) mRNA levels in bulk cultured primary tonsillar fibroblast of OSA patients stimulated with the indicated recombinant proteins for 48 h and measured by qRT-PCR. **d**, Fold changes of *PI16* mRNA levels after in vitro expansion of sorted *PI16*+ RCs from human tonsils and following stimulation with TGF-β1 for 48 h. **b**–**d**, Box plots with whiskers showing the minimum and maximum values. Horizontal lines indicate the median and boxes represent 0.25−0.75 percentiles (**b**, **c**) q values are derived from Tukey's test following Kruskal-Wallis test and using Benjamini, Krieger and Yekutieli correction to control the false discovery rate. (**d**) Two-sided Mann Whitney test was used to test for significant differences. In **a**, data represent 3,450 *CXCL13*+ cells and 56,887 immune cells sampled from n = 4 patients for human lymph nodes and n = 4 patients for human palatine tonsils. In **b**, **c**, cells from n = 8 patients were cultivated and processed in 2 independent experiments. In **d**, cells from n = 5 patients were cultivated and processed in 2 independent experiments.

**Extended Data Table 1 | Patient characteristics**

| Patient number | Sex[a] | Age[b] (years) | Diagnosis | Data acquired |
|---|---|---|---|---|
| LN 01 | M | 39.1 | Pronounced chronic hyperplastic tonsillitis with focal purulent ulcerative tonsillitis | scRNAseq stromal cells, T and B cells |
| LN 02 | M | 36.8 | Cervical cyst | scRNAseq stromal cells |
| LN 03 | F | 38.1 | Pleomorphic adenoma | scRNAseq stromal cells, T, B and myeloid cells |
| LN 04 | F | 40.0 | Basal cell adenoma | scRNAseq stromal cells, T, B and myeloid cells |
| LN 05 | F | 65.3 | Sialolithiasis of the submandibular gland | Histology |
| LN 06 | M | 35 | Cervical cyst | Histology |
| LN 07 | M | 39.3 | Warthin tumor | Histology |
| LN 08 | M | 65.8 | Warthin tumor | Histology |
| Tonsil 01 | M | 54.6 | OSA / Tonsil Hyperplasia | scRNAseq stromal cells / in vitro bulk fibroblast stimulation |
| Tonsil 02 | M | 30.4 | OSA / Tonsil Hyperplasia | scRNAseq stromal cells / in vitro bulk fibroblast stimulation |
| Tonsil 03 | M | 37.3 | OSA / Tonsil Hyperplasia | scRNAseq stromal cells, T, B and myeloid cells |
| Tonsil 04 | M | 39.1 | Pronounced chronic hyperplastic tonsillitis with focal purulent ulcerative tonsillitis | scRNAseq stromal cells, T and B cells |
| Tonsil 05 | F | 37.9 | OSA / Tonsil Hyperplasia | Histology |
| Tonsil 06 | M | 28.8 | OSA / Tonsil Hyperplasia | Histology |
| Tonsil 07 | M | 8.9 | OSA / Tonsil Hyperplasia | in vitro bulk fibroblast stimulation |
| Tonsil 08 | M | 5.7 | OSA / Tonsil Hyperplasia | in vitro bulk fibroblast stimulation |
| Tonsil 09 | unknown | unknown | OSA / Tonsil Hyperplasia | in vitro bulk fibroblast stimulation |
| Tonsil 10 | F | 6.5 | OSA / Tonsil Hyperplasia | in vitro bulk fibroblast stimulation |
| Tonsil 11 | F | 42.3 | OSA / Tonsil Hyperplasia | in vitro bulk fibroblast stimulation |
| Tonsil 12 | M | 42.4 | OSA / Tonsil Hyperplasia | in vitro bulk fibroblast stimulation |
| Tonsil 13 | F | 24.1 | OSA / Tonsil Hyperplasia | in vitro stimulation PI16$^+$RC |
| Tonsil 14 | M | 53.8 | OSA / Tonsil Hyperplasia | in vitro stimulation PI16$^+$RC |
| Tonsil 15 | F | 28.3 | Tonsillitis (PTA) one side | in vitro stimulation PI16$^+$RC |
| Tonsil 16 | M | 24.1 | Tonsillitis | in vitro stimulation PI16$^+$RC |
| Tonsil 17 | M | 26.0 | Tonsillitis | in vitro stimulation PI16$^+$RC |

[a] Male to female ratio of 1.67:1 for tissues from LN donors, and 1:2.2 for tonsil donors
[b] Mean age of LN donors: $44.9 \pm 12.8$ years; mean age of tonsil donors: $30.6 \pm 1$

# Reporting Summary

## Statistics

For all statistical analyses, confirm that the following items are present in the figure legend, table legend, main text, or Methods section.

| n/a | Confirmed | |
|---|---|---|
| ☐ | ☒ | The exact sample size (*n*) for each experimental group/condition, given as a discrete number and unit of measurement |
| ☐ | ☒ | A statement on whether measurements were taken from distinct samples or whether the same sample was measured repeatedly |
| ☐ | ☒ | The statistical test(s) used AND whether they are one- or two-sided *Only common tests should be described solely by name; describe more complex techniques in the Methods section.* |
| ☐ | ☒ | A description of all covariates tested |
| ☐ | ☒ | A description of any assumptions or corrections, such as tests of normality and adjustment for multiple comparisons |
| ☐ | ☒ | A full description of the statistical parameters including central tendency (e.g. means) or other basic estimates (e.g. regression coefficient) AND variation (e.g. standard deviation) or associated estimates of uncertainty (e.g. confidence intervals) |
| ☐ | ☒ | For null hypothesis testing, the test statistic (e.g. *F*, *t*, *r*) with confidence intervals, effect sizes, degrees of freedom and *P* value noted *Give P values as exact values whenever suitable.* |
| ☒ | ☐ | For Bayesian analysis, information on the choice of priors and Markov chain Monte Carlo settings |
| ☒ | ☐ | For hierarchical and complex designs, identification of the appropriate level for tests and full reporting of outcomes |
| ☐ | ☒ | Estimates of effect sizes (e.g. Cohen's *d*, Pearson's *r*), indicating how they were calculated |

*Our web collection on statistics for biologists contains articles on many of the points above.*

## Software and code

Policy information about availability of computer code

| Data collection | FACSDiva (BD Biosciences, v8.0.1 and v9.0.1), FACSChorus (BD Biosciences, v1.3), ZEN blue (Zeiss, v3.3) |
|---|---|
| Data analysis | FlowJo (Treestar Inc., v10), R (v.4.0.0), CellRanger (v3.0.2), scater R/Bioconductor package (v.1.16.0), Seurat R package (v.4.0.1), clusterProfiler R/Bioconductor package (v.3.15.3), variancePartition R package (v.1.22.0), muscat R/Bioconductor package (v.1.6.0), CellPhone-DB (v.2.1.7), python v.3.7.0, Imaris (v9), Space Ranger software (v.1.3.1), Loupe Browser (v.6.2.0), SpatialExperiment R/Bioconductor package (v.1.6.1), spacexr R package (v.2.0.1). Code used for data analysis in this project is available at github (https://github.com/mluetge/CrossSLO_BRC_CXCL13) |

For manuscripts utilizing custom algorithms or software that are central to the research but not yet described in published literature, software must be made available to editors and reviewers. We strongly encourage code deposition in a community repository (e.g. GitHub). See the Nature Portfolio guidelines for submitting code & software for further information.

## Data

Policy information about availability of data

All manuscripts must include a data availability statement. This statement should provide the following information, where applicable:
- Accession codes, unique identifiers, or web links for publicly available datasets
- A description of any restrictions on data availability
- For clinical datasets or third party data, please ensure that the statement adheres to our policy

Ensembl GRCm38.9 and GRCh38.9 were used as reference genomes to build the indexes. The scRNA-seq and spatial transcriptomics data generated in this study have been deposited in the BioStudies database (www.ebi.ac.uk/biostudies/). Mouse scRNAseq data is available under accession code E-MTAB-11738, human lymph node data is available under E-MTAB-11710 and data from human palatine tonsils is available under E-MTAB-11715. Spatial transcriptomics data is available

# Field-specific reporting

Please select the one below that is the best fit for your research. If you are not sure, read the appropriate sections before making your selection.

☒ Life sciences ☐ Behavioural & social sciences ☐ Ecological, evolutionary & environmental sciences

For a reference copy of the document with all sections, see nature.com/documents/nr-reporting-summary-flat.pdf

# Life sciences study design

All studies must disclose on these points even when the disclosure is negative.

| | |
|---|---|
| Sample size | No sample-size calculation was performed. Sample sizes were determined to be adequate based on the reproducibility between independent experiments and adequate cell numbers of each subset in the scRNA-seq data to run comparative analyses. Sample sizes for both single cell and experimental studies were based on our experience and common practise in the field (Nat Immunol. 2020 Jun;21(6):649-659). |
| Data exclusions | No data points were excluded. |
| Replication | For analysis of the performed scRNA-seq experiments no batch correction needed to be applied for any of the samples. All attempts at replication were successful. For all experimental analyses, all attempts at replication were successful. |
| Randomization | Randomization and control of covariants was not relevant in the setting of this exploratory study. |
| Blinding | Blinding was not performed since data analysis was explorative. |

# Reporting for specific materials, systems and methods

We require information from authors about some types of materials, experimental systems and methods used in many studies. Here, indicate whether each material, system or method listed is relevant to your study. If you are not sure if a list item applies to your research, read the appropriate section before selecting a response.

## Materials & experimental systems

| n/a | Involved in the study |
|---|---|
| ☐ | ☒ Antibodies |
| ☐ | ☒ Eukaryotic cell lines |
| ☒ | ☐ Palaeontology and archaeology |
| ☐ | ☒ Animals and other organisms |
| ☐ | ☒ Human research participants |
| ☒ | ☐ Clinical data |
| ☒ | ☐ Dual use research of concern |

## Methods

| n/a | Involved in the study |
|---|---|
| ☒ | ☐ ChIP-seq |
| ☐ | ☒ Flow cytometry |
| ☒ | ☐ MRI-based neuroimaging |

# Antibodies

| | |
|---|---|
| Antibodies used | Histology antibodies:<br>Anti-GFP polyclonal chicken (Aves Labs Inc., Cat#: GFP-1020, Lot#: GFP879484, 1:1000)<br>Anti-DsRed polyclonal rabbit (Takara Bio Clontech, Cat#: 632496, Lot#: 1805060, 1:1000)<br>Anti-human/mouse B220 eFluor450 (Thermo Scientific, Cat#: 48-0452-82, clone: RA3-6B2, Lot#: 2195593, 1:200)<br>Anti-mouse Lyve1 eFluor660 (Thermo Scientific, Cat#: 50-0443-82, clone:ALY-7, Lot#: 2205461, 1:200)<br>Anti-mouse CD4 AlexaFluor488 (BioLegend, Cat#: 100529, clone: RM4-5, Lot#: B243360 ,1:200)<br>Anti-mouse F4/80 AlexaFluor647 (BioLegend, Cat#: 123122, clone: BM8, Lot#: B265213, 1:200)<br>Anti-mouse PDPN syrian hamster (BioLegend, Cat#: 127402, clone: 8.1.1, Lot#: B228668, 1:300)<br>Anti-mouse CD21/35 APC (BioLegend, Cat#: 123412, clone: 7E9, Lot#: B256895, 1:500)<br>Anti-mouse MAdCAM1 Biotin (BioLegend, Cat#: 120706, clone: MECA-367, Lot#: B187519, 1:200)<br>Anti-mouse CD157 APC (BioLegend, Cat#: 140208, clone: BP-3, Lot#: B213863, 1:100)<br>Anti-mouse CD31 AlexaFluor647 (BioLegend, Cat#: 102516, clone: MEC13.3, Lot#: B308659, 1:200)<br>Anti-mouse PNAd Biotin (BioLegend, Cat#: 120804, clone: MECA-79, Lot#: B177144, 1:200)<br>Anti-mouse CD326/EPCAM AlexaFluor488 (BioLegend, Cat#: 118210, clone: G8.8, Lot#: B285223, 1:200)<br>Anti-mouse CXCL13 Biotin polyclonal goat (R&D Systems ,Cat#: BAF470, Lot#: DAD0314121, 1:100)<br>Anti-mouse CCL21 Biotin polyclonal goat (R&D Systems, Cat#: BAF457, Lot#: BEO0819071, 1:100)<br>Anti-mouse TRANCE/RANKL polyclonal goat (R&D Systems, Cat#: AF462, Lot#: CKN0320092, 1:200)<br>Anti-human / mouse SMA Cy3 (Sigma, Cat#: C6198, clone:1A4, Lot#: 0000116745, 1:400)<br>Anti-human CD3 polyclonal rabbit (Dako, Cat#: A045201-2, Lot#: 20061852, 1:200)<br>Anti-human CD20 AlexaFluor488 (Thermo Scientific, Cat#: 53-0202-82, clone: L26, Lot#: 2210882, 1:200) |

Anti-human PDPN (Thermo Scientific, Cat#: 14-9381-82, clone: NZ-1.3, Lot#: 2400405, 1:200)
Anti-human Clusterin (BD Biosciences, Cat#: 552886, clone: E5, Lot#: 9346561, 1:200)
Anti-human CXCL13 (R&D Systems, Cat#: MAB801, clone: 53610, Lot#: BJT0819081, 1:100)
Anti-human CCL19 polyclonal rabbit (Abcam, Cat#: ab221704, Lot#: GR3409392-1, 1:100)
Anti-Chicken AlexaFluor488 polyclonal (Jackson ImmunoResearch, Cat#: 703-545-155, Lot#: 151901, 1:1000)
Anti-Goat AlexaFluor488 polyclonal (Jackson ImmunoResearch, Cat#: 705-545-003, Lot#: 148783, 1:1000)
Anti-Goat AlexaFluor647 polyclonal (Jackson ImmunoResearch, Cat#: 705-605-003, Lot#: 153846, 1:1000)
Anti-Mouse Cy3 polyclonal (Jackson ImmunoResearch, Cat#: 715-165-150, Lot#: 155993, 1:1000)
Anti-Rabbit AlexaFluor488 (Jackson ImmunoResearch, Cat#: 711-545-152, Lot#: 158217, 1:1000)
Anti-Rabbit Cy3 polyclonal (Jackson ImmunoResearch, Cat#: 711-165-152, Lot#: 157936, 1:1000)
Anti-Syrian hamster AlexaFluor488 (Jackson ImmunoResearch, Cat#: 107-545-142, Lot#:150847, 1:100)
Anti-Rat AlexaFluor647 (Jackson ImmunoResearch, Cat#: 712-606-150, Lot#: 150018, 1:1000)
Streptavidin AlexaFluor488 (Jackson ImmunoResearch, Cat#: 016-540-084, Lot#: 138230, 1:1000)
Streptavidin Cy3 (Jackson ImmunoResearch, Cat#: 016-160-084, Lot#: 141873, 1:1000)
Streptavidin-AlexaFluor647 (Jackson ImmunoResearch, Cat#: 016-600-984, Lot#: 141873, 1:1000)

Flow cytometry and cell sorting antibodies:
Anti-mouse CD157 BV786 (BD Biosciences, Cat#: 741012, clone: BP-3, Lot#: 1047515, 1:200)
Anti-mouse CD45.2 BV605 (BioLegend, Cat#: 109841, clone: 104, Lot#: B310111, 1:200)
Anti-mouse CD31 AlexaFluor647 (BioLegend, Cat#: 102516, clone: MEC13.3, Lot#: B308659, 1:200)
Anti-mouse PDPN PE-Cy7 (BioLegend, Cat#: 127412, clone: 36899, Lot#: B310444, 1:200)
Anti-mouse CD21/35 Pacific Blue (BioLegend, Cat#: 123414, clone: 7E9, Lot#: B294084, 1:200)
Anti-mouse MAdCAM1 Biotin (BioLegend, Cat#: 120706, clone: MECA-367, Lot#: B187519, 1:200)
Anti-mouse SCA1 APC-Cy7 (BioLegend, Cat#: 108126, clone: D7, Lot#: B253002, 1:200)
Anti-mouse CD19 APC-Cy7 (BioLegend, Cat#: 115530, clone: 6D5, Lot#: B290859, 1:200)
Anti-mouse CD38 PE (BioLegend, Cat#: 102708, clone: 90, Lot#: 2209694, 1:200)
Anti-mouse/human GL7 AlexaFluor488 (BioLegend, Cat#: 144612, clone: GL-7, Lot#: B256896, 1:200)
Anti-mouse CD4 PE (BioLegend, Cat#: 116006, clone: RM4-5, Lot#: B186627, 1:200)
Anti-mouse CD11b AlexaFluor647 (BioLegend, Cat#: 101218, clone: M1/70, Lot#: B236178, 1:200)
Anti-human CD45 FITC (BD Biosciences, Cat#: 555482, clone: H130, Lot#: 8120939, 1:200)
Anti-human PDPN PE (Thermo Scientific, Cat#: 12-9381-42, clone: NZ-1.3, Lot#: 4332768, 1:200)
Anti-human CD31 APC (Thermo Scientific, Cat#: 17-0319-42, clone: WM59, Lot#: 1976592, 1:200)
Anti-human EPCAM FITC (BioLegend, Cat#: 324203, clone: 9C4, Lot#: B261791, 1:200)
Anti-human CD14 PE-Cy7 (BioLegend, Cat#: 301813, clone: MSE2, Lot#: B231081, 1:200)
Anti-human CD3 FITC (BioLegend, Cat#: 300440, clone: UCHT1, Lot#: B279209, 1:200)
Anti-human CD19 APC/Fire 750 (BioLegend, Cat#: 302258, clone: HIB19, Lot#: B242981, 1:200)
Anti-human CD45 PeCy7 (Thermo Scientific, Cat#:25-0459-42, Clone: HI30, Lot#: 2079970, 1:200)
Anti-human CD31 Biotin (Thermo Scientific, Cat#:13-0319-82, Clone: WM59, Lot#: 1994108, 1:200)
Anti-human CD235a PeCy7 (BioLegend, Cat#:349112, Clone: HI264, Lot#: B274230, 1:200)
Anti-human CD34 FITC (BioLegend, Cat#:343504, Clone: 581, Lot#: B356958, 1:200)
Streptavidin-BV711 (BioLegend, Cat#: 405241, Lot#: B332218, 1:1000)

Validation | All antibodies came from commercial vendors, and were validated by the manufacturers on their official website. For stainings that used a combination of primary and secondary antibodies, each primary antibody was additionally validated by performing control stains using the secondary antibody alone to ensure a specific signal.

# Eukaryotic cell lines

Policy information about cell lines

Cell line source(s) | Vero ATCC CCL-18

Authentication | The functionality of the cell line was confirmed in our labortatory, as demonstrated by the ability to be infected by VSV.

Mycoplasma contamination | Vero cell lines tested negative for mycoplama contamination in our lab.

Commonly misidentified lines (See ICLAC register) | No misidentified lines were used

# Animals and other organisms

Policy information about studies involving animals; ARRIVE guidelines recommended for reporting animal research

Laboratory animals | Experiments were performed with 6 to 10 week-old mice (males and females). All animals were housed in individually ventilated cages under conventional specific pathogen-free conditions, maintaining a 12 hour light/dark cycle, 22°C ambient temperature and 45/50% humidity. All mouse strains were on a C57BL/6N Charles River genetic background. The C57BL/6N-Tg(Cxcl13-Cre/TdTomato) x R26-EYFP strain was described previously. C57BL/6N-Tg(Cxcl13-Cre)723Biat x B6.129-Il6tm1Jho (Cxcl13-Cre Il6fl/fl) mice were generated by crossing Cxcl13-Cre mice with B6.129-Il6tm1Jho (Il6fl/fl) mice that were described before.

Wild animals | none

Field-collected samples | none

| Ethics oversight | Experiments were performed with 6- to 10-week-old mice (males and females) and were in accordance with Swiss federal and cantonal guidelines (Tierschutzgesetz) under permissions SG/26/2020 granted by the Veterinary Office of the Canton of St. Gallen. |
|---|---|

Note that full information on the approval of the study protocol must also be provided in the manuscript.

# Human research participants

Policy information about studies involving human research participants

| Population characteristics | Detailed information is listed in Extended data Table 1 "Patient characteristics" |
|---|---|
| Recruitment | Tonsil samples were collected from adult patients suffering from obstructive sleep apnea (OSA) due to hyperplastic tonsils that underwent routine tonsillectomy at the Kantonsspital St. Gallen. Decision to perform surgery was made after clinical assessment by attending ENT physicians. Patient material was used following provision of informed consent.<br>Lymph node samples were collected from adult patients with benign LN swelling conditions that underwent routine lateral parotidectomy or transcervical excision of a cervical cyst at the Kantonsspital St. Gallen. Decision to perform surgery was made after clinical assessment by attending ENT physicians. Patient material was used following provision of informed consent. |
| Ethics oversight | Ethikkommission Ostschweiz (EKOS), Kantonsspital, Haus 37, 9007 St. Gallen |

Note that full information on the approval of the study protocol must also be provided in the manuscript.

# Flow Cytometry

## Plots

Confirm that:

☒ The axis labels state the marker and fluorochrome used (e.g. CD4-FITC).

☒ The axis scales are clearly visible. Include numbers along axes only for bottom left plot of group (a 'group' is an analysis of identical markers).

☒ All plots are contour plots with outliers or pseudocolor plots.

☒ A numerical value for number of cells or percentage (with statistics) is provided.

## Methodology

| Sample preparation | A description of the sample preparation for flow cytometry and FACS sorting is detailed in the methods section. |
|---|---|
| Instrument | LSR Fortessa BD Biosciences, FACS Melody BD Biosciences |
| Software | FACSDiva and FACSChorus (both from BD) were used for data acquisition. FlowJO v10 (Treestar inc.) was used to analyze the data. |
| Cell population abundance | A test sample was prepared for testing the purity of sorted cells. The purity of the post-sort fraction was determined by flow cytometry using the LSR Fortessa 2. |
| Gating strategy | For all flow cytometric analyses, cells were first gated on FSC/SSC to exclude cell debris following by FSC-A/FSC-H to exclude doublets. Dead cells were excluded from analysis by gating on cells staining negative for a viability dye. A detailed gating strategy for stromal and immune cell populations is provided in the Extended data. |

☒ Tick this box to confirm that a figure exemplifying the gating strategy is provided in the Supplementary Information.

