## [Peer Review File · Nature Immunology]

Peer Review Information

Journal: Nature Immunology

Manuscript Title: Conserved stromal-immune cell circuits secure B cell homeostasis and function

Corresponding author name(s): Professor Burkhard Ludewig, Dr Natalia Pikor

Reviewer Comments & Decisions:

Decision Letter, initial version:
--

30th May 2022

Dear Burkhard,

Thank you for providing a point-by-point response to the referees' concerns on your manuscript entitled "Conserved stromal-immune cell circuits secure B cell homeostasis and function". It was good to discuss the comments and proposed revision plans with you and your students at the recent Aegean conference. (Apologies for the delay in getting back to you with the formal decision letter as I just catching up on correspondence after doing lab visits in Heraklion today). As noted, although we cannot accept the current version of the manuscript, we are very much interested in a revised version that addresses the referees' comments.

We invite you to submit a substantially revised manuscript, however please bear in mind that we will be reluctant to approach the referees again in the absence of major revisions.

Specifically, the revision should include new experiments to address:

- (1) sort tdTomato+ CXC13-Cre cells and perform in vitro stimulation with predicted niche ligands.
- (2) assay for BRC subset identity & BRC signaling pathways
- (3) validate predicted novel functions played by niche-specific BRCs, including IL-6 crosstalk between Pi16+ BRCs with Tfh cells, & impacts on humoral immunity.

Please include the additional textual clarifications as indicated in your response letter.

When you revise your manuscript, please take into account all reviewer and editor comments, please highlight all changes in the manuscript text file in Microsoft Word format.

* If you have not done so already please begin to revise your manuscript so that it conforms to our Article format instructions at <http://www.nature.com/ni/authors/index.html>. Refer also to any guidelines provided in this letter.

The Reporting Summary can be found here:

When submitting the revised version of your manuscript, please pay close attention to our [href="https://www.nature.com/nature-portfolio/editorial-policies/image-integrity">Digital Image Integrity Guidelines. and to the following points below:](https://www.nature.com/nature-portfolio/editorial-policies/image-integrity)

[REDACTED]

If you wish to submit a suitably revised manuscript we would hope to receive it within 6 months. If you cannot send it within this time, please let us know. We will be happy to consider your revision so long as nothing similar has been accepted for publication at Nature Immunology or published elsewhere.

Nature Immunology is committed to improving transparency in authorship. As part of our efforts in this direction, we are now requesting that all authors identified as 'corresponding author' on published

papers create and link their Open Researcher and Contributor Identifier (ORCID) with their account on the Manuscript Tracking System (MTS), prior to acceptance. ORCID helps the scientific community achieve unambiguous attribution of all scholarly contributions. You can create and link your ORCID from the home page of the MTS by clicking on 'Modify my Springer Nature account'. For more information please visit www.springernature.com/orcid.

Thank you for the opportunity to review your work.

Kind regards,

Laurie

Laurie A. Dempsey, Ph.D.
Senior Editor
Nature Immunology
l.dempsey@us.nature.com
ORCID: 0000-0002-3304-796X

Referee expertise:

Referee #1: B cells

Referee #2: Germinal centers

Referee #3: Germinal centers

Reviewers' Comments:

Reviewer #1:

Remarks to the Author:

This study takes advantage of CXCL13 reporter mice previously generated by this lab to compare B zone reticular cells (BRCs) across spleen, LN and PP. Initial data confirm expected in situ expression data for CXCL13 in these tissues while also providing improved sensitivity that allows weakly expressed cells to be better detected. Flow cytometric comparison across the tissues clarifies the relationship between expression of the FRC marker, PDPN, and CXCL13+ cells in the different tissues, as well as several other FRC markers. Droplet based 10x scRNAseq analysis of CXCL13-reporter+ cells then allowed deeper comparison of stromal cells across the three lymphoid tissues, and in spleen and LN at day 15 after VSV infection. As expected from the many prior anatomical studies, the main stromal cell types were conserved across the three tissue types. A new finding is the presence of Pi16+ RCs across the lymphoid tissues, and their localization by IF proximal to vessels. MDS plots show the stroma are imprinted by tissue of origin, extending earlier studies that have shown this for example through reciprocal transplants of mesenteric and peripheral LNs. The imprinting is found to

connect with distinct processes that relate to known properties of the different organs. scRNAseq data are then collected on immune cells and inferences are made, for example with CellPhoneDB, about conserved predicted BRC-immune cell interactions. Leukocyte-derived factors that promote BRC identity are inferred from these data, including established ligand-receptor relationships and the suggestion of some yet to be studied interactions such as macrophage TGF β engaging TGFBR on Pi16+ RC. Finally, scRNAseq data are collected from human LNs (n=4) and tonsils (n=4) and genes expressed by CXCL13+ BRC are detailed. These data show good species conservation of the main BRC subsets. They also allow conserved potential immune cell-BRC communications to be highlighted.

Overall, this is considered a well performed descriptive study. The extensive stromal cell data provide a further resource above the stromal cell data from each of these tissues that has already been reported by this and other groups. The quality of the IF images is generally excellent, and this helps extend some of the insights (such as identification of the Pi16+ subset) from sequence space into anatomical space. However:

1. My main concern in the context of consideration for NI is whether the study represents a sufficient advance. While the careful acquisition and curation of sequencing data is important for the field, for mouse-based studies it is typical to pursue one or more of the inferences from the data at a mechanistic level. This seems particularly important here as several of the central observations are otherwise confirmatory. For example, are the Pi16+ RC affected by interference with TGF β signaling? Have any of the newly inferred communications been experimentally verified?
2. The scRNAseq data, as with past data studies by this group, has been deposited as primary data as permitted by the ArrayExpress database. However, the more widely used GEO database requires processed data to be deposited. It can be challenging for labs that lack advanced computing power to download the raw data and run the 10x cellranger pipeline themselves, especially if they are just wanting to quickly query the data. It is recommended that processed data are also deposited. Moreover, many studies that involve largely scRNAseq data acquisition and analysis include a user-friendly portal to facilitate data querying. Such a portal is likely to increase the utility and citation of the study. (It should be noted that the human palatine tonsil data reported here as available under E-MTAB-11715 is the same dataset analyzed in the accompanying manuscript by De Martin et al., which also indicates depositing the dataset).

Reviewer #2:

Remarks to the Author:

Pikor and colleagues utilise their Cxcl13-cre-TdTomato strain to characterise CXCL13 expressing cells in lymph nodes, spleen and peyer's patches by scRNAseq and flow cytometry, akin to their 2020 Nat Immunol paper that reported a similar analysis of lymph node CXCL13-expressing cells. By mining these data, they are able to see that there are populations of CXCL13-expressing cells that are common amongst lymphoid organs, including a previously undescribed population of Pi16-expressing stromal cells. In addition, CXCL13 is expressed by many different cell types that are unique to the different organs. The scRNAseq data are used to identify pathways that are enriched in different CXCL13+ stromal cell types. Further scRNAseq of hematopoietic cells followed by interaction analyses identified putative interactions between cell types. A similar scRNAseq analysis was conducted on human lymph nodes, with similar results, highlighting conserved cell types between species.

Reviewer #3:

Remarks to the Author:

The manuscript by Lütge et al investigated CXCL13-expressing reticular cell subsets in different secondary lymphoid organs, as well as by comparing samples from mice to humans. Overall findings are that B cell-interacting reticular cell (BRC) subsets shared similarities across different SLOs in terms of gene expression and topology, but there were tissue-specific gene signatures for each subset. Shared BRC subsets in different SLOs had canonical subset markers, as well as gene expression programs unique to each SLO, based on GO term analysis. Conclusions are drawn about BRC-immune cell interactions and that BRCs govern humoral immune responses.

Major comments

The experiments are technically well done, the manuscript is clear and has appropriately referenced other papers in this field of study. The major approach of the study is to describe single-cell gene expression signatures with a number of different bioinformatical methods. Using this intricate approach, the study makes a number of novel, useful observations about BRC subsets in SLOs, in both mice and humans.

However, the major gap in this study is the lack of follow-on experimentation from initial scRNAseq in the different contexts. The authors make a number of broad mechanistic conclusions but do not undertake further experimental exploration of testable hypotheses. For example: subset-specific genes – can you modulate tissue-specific subset identity or function by modulating gene expression? Will it change the outcome of local responses? Furthermore, the language used in the conclusions is not completely supported by the data at hand and would be more appropriate when mechanistic experiments are performed (eg 'no functional overlap', 'bidirectional molecular circuits', 'feedforward circuits', '...suggestive of conserved BRC subset-specific interactions steering regionalized niches in and around the B cell follicle'.)

BRC-immune cell interactions were predicted from droplet-based scRNAseq and the significance of these interactions in terms of humoral immunity was inferred. This is another area where further mechanistic insight could be generated. For example, the authors infer improved B cell immunity based on enrichment of specific signalling pathways. If inferring role of BRCs in modulating GC responses, this be demonstrated with conditional KO studies. If BRCs don't express these receptors are GC responses altered or abrogated? Is the 'efficacy of humoral immunity' altered when these pathways are dysregulated?

Minor comments

There were a few sentences that weren't clear in its intent or need references, eg

- 1) Lines 54-57 appears to be conflating germinal centers with BRCs
- 2) When discussing 'DZ FDCs' – can the authors please provide references.

Author Rebuttal to Initial comments

See Inserted PDF

Dear Laurie, Dear Editors of Nature Immunology,

We would like to thank the reviewers for the positive feedback and constructive suggestions and the editors for the opportunity to revise our manuscript entitled “Conserved stromal-immune cell circuits secure B cell homeostasis and function”. We have productively addressed all of the reviewers’ comments in the point-by-point reply below. The amended data is summarized in the text below and in the revised manuscript, which includes:

- a new Figure 5 and Extended Data Figure 5 consisting of in vitro and in vivo experimental validation experiments of the biological function of newly inferred BRC differentiation and activation factors, as well as genetic validation of the role for a BRC niche factor to steer humoral immunity,
- an updated Extended Data Figure 8 containing experimental validation of BRC differentiation and activation factors in human tonsillar FRCs.

Reviewer 1:

This study takes advantage of CXCL13 reporter mice previously generated by this lab to compare B zone reticular cells (BRCs) across spleen, LN and PP. Initial data confirm expected in situ expression data for CXCL13 in these tissues while also providing improved sensitivity that allows weakly expressed cells to be better detected. Flow cytometric comparison across the tissues clarifies the relationship between expression of the FRC marker, PDPN, and CXCL13+ cells in the different tissues, as well as several other FRC markers. Droplet based 10x scRNAseq analysis of CXCL13-reporter+ cells then allowed deeper comparison of stromal cells across the three lymphoid tissues, and in spleen and LN at day 15 after VSV infection. As expected from the many prior anatomical studies, the main stromal cell types were conserved across the three tissue types. A new finding is the presence of Pi16+ RCs across the lymphoid tissues, and their localization by IF proximal to vessels. MDS plots show the stroma are imprinted by tissue of origin, extending earlier studies that have shown this for example through reciprocal transplants of mesenteric and peripheral LNs. The imprinting is found to connect with distinct processes that relate to known properties of the different organs. scRNAseq data are then collected on immune cells and inferences are made, for example with CellPhoneDB, about conserved predicted BRC-immune cell interactions. Leukocyte-derived factors that promote BRC identity are inferred from these data, including established ligand-receptor relationships and the suggestion of some yet to be studied interactions such as macrophage TGFb engaging TGFBR on Pi16+ RC. Finally, scRNAseq data are collected from human LNs (n=4) and tonsils (n=4) and genes expressed by CXCL13+ BRC are detailed. These data show good species conservation of the main BRC subsets. They also allow conserved potential immune cell-BRC communications to be highlighted.

Overall, this is considered a well performed descriptive study. The extensive stromal cell data provide a further resource above the stromal cell data from each of these tissues that has already been reported by this and other groups. The quality of the IF images is generally excellent, and this helps extend some of the insights (such as identification of the Pi16+ subset) from sequence space into anatomical space. However: 1. My main concern in the context of consideration for NI is whether the study represents a sufficient advance. While the careful acquisition and curation of sequencing data is important for the field, for mouse-based studies it is typical to pursue one or more of the inferences from the data at a mechanistic level. This seems particularly important here as several of the central observations are otherwise confirmatory. For example, are the Pi16+ RC affected by interference with TGFb signaling? Have any of the newly inferred communications been experimentally verified?

We thank the reviewer for this comment. Indeed, in our study we reasoned that the confirmation of known BRC niche factors and recapitulation of the identified receptor-ligand pairs across murine and human lymphoid tissues supports the validity of the stringent bioinformatics pipeline applied. Nonetheless, we agree with the reviewer that experimental validation will significantly strengthen our conclusions.

To provide mechanistic evidence for the newly inferred interactions, we applied several strategies in parallel. Firstly, we stimulated lymph node fibroblasts *in vitro* with the identified differentiation and activation factors. Specifically, we isolated fibroblasts from lymph nodes of *Cxcl13-Cre/TdTomato* mice and performed *in vitro* stimulations with recombinant murine IL4, VEGF-B, IL-1 β , TGF β and Progranulin (PRGN), and analyzed *Cxcl13-Cre/TdTomato* expression 48 hours later by flow cytometry. Indeed, we found that IL-4, VEGF-B, TGF β and PGRN, but not IL-1 β , induced TdTomato (CXCL13) expression validating their function as BRC differentiation cues. These data are now included in the manuscript in the new Figure 5e-f.

To examine subset-specific BRC interactions, we applied complementary approaches to overcome technical limitations in the yield and cultivation capacity of rare BRC subsets such as FDCs and PI16⁺ RCs. As FDCs were particularly challenging to culture, we assessed FDC-specific differentiation and activation in an *in vivo* setting. To test for FDC subset specification and function, we injected the identified BRC differentiation and activation factors, recombinant murine VEGF-B, IL-4, or IL-1 β subcutaneously in the flank of *Cxcl13-Cre/TdTomato* mice and assessed the frequency, maturation and function of FDCs 48 hours after three consecutive days of stimulation. IL-1 β was found to induce an increase in FDC frequency, while IL-4 induced the expression of the activation marker and FDC niche factor ICAM (new Figure 5h-i). To further demonstrate that activation was associated with FDC function, we assessed immune complex retention following *in vivo* stimulation with IL4, IL1- β or VEGF-B compared to PBS. For this, mice were injected subcutaneously with either one of the cytokines or PBS on three consecutive days with immune complexes (anti-PE(mouse IgG)⁺ PE) being added to the third injection. Immune complex retention by inguinal lymph node FDCs was analyzed 48 hours after the last immunization using flow cytometry. Confirming the predicted effect on FDC subset identity, we found that IL-4 injection significantly increased immune complex retention compared to PBS treatment (new Figure 5j-l). Collectively, these results reveal the biological mechanisms supported by the identified interactions, wherein VEGF-B induces *Cxcl13* expression, IL-1 β promotes FDC expansion and IL-4 supports FDC niche factor production and function. Notably, *in vitro* stimulation of human tonsillar fibroblasts with IL-1 β was observed to induce the expression of the FDC-specific transcription factor *SOX9*, reinforcing the role of IL-1b as an FDC differentiation factor (new Extended Data Figure 8b).

These data are now included in the manuscript in the new Figure 5 and Extended Data Figures 5 and 8.

In addition, to further substantiate data in the manuscript for the roles of IL-4 and IL-1 β as activation and differentiation factors, we have included the following data for the reviewer's attention. Following *in vivo* stimulations, both IL-4 and IL-1 β were found to induce the expression of the activation marker SCA-1 on FDCs (Reviewer Figure 1).

Reviewer Figure 1. Experimental validation of FDC subset specification factors. a) Quantification of the mean fluorescence intensity (MFI) of SCA-1 by CD21/35⁺ *Cxcl13-Cre/TdTomato*⁺ cells *in vivo* stimulation with the indicated maturation factors.

To provide mechanistic evidence for the interaction and biological function of the identified factors that affect PI16⁺ RCs, we examined the transcriptional reprogramming of murine lymph node and human tonsillar fibroblasts. Stimulation of murine lymph node fibroblasts with PRGN and TGFβ caused a reduction in the amount of IL-6 in culture supernatants (new Figure 5g). Furthermore, in the human setting, stimulation of tonsillar fibroblasts with TGFβ induced increase PI16 expression. As bulk cultures are not necessarily subset-specific, we additionally sorted PI16⁺ RCs from human tonsils and stimulated them with the predicted PI16⁺ specification factors. We elected for this approach given the larger yield of fibroblasts from this tissue and the lack of surface markers for flow cytometric separation in the murine setting. As described by De Martin et al. CD34⁺ PDPN⁺ CD31⁻ UEA1⁻ cells demarcate PI16⁺ RCs in human tonsils (see accompanying manuscript). Sorted tonsillar fibroblasts were stimulated with the recombinant human proteins TGFβ, PRGN and Growth Hormone (GH). Stimulation of both bulk and sorted PI16⁺ RC cultures induced the expression of *PI16*. This data is now shown in the new Extended Data Figure 8c-d.

For the attention of the reviewers, we additionally provide data showing that murine lymph node fibroblasts stimulated with TGFβ induce *Pi16* expression, consistent with the observed role for TGFβ to promote PI16⁺ RC specification in human tonsillar fibroblasts. We additionally tested whether the predicted PI16⁺ RC activation factors affected the expression of the PI16⁺ RC niche factors SEMA3C and IL-6. TGFβ1 induced the expression of the predicted subset-specific niche factor *SEMA3C* in bulk human tonsillar tissue cultures, and in 2 out of 5 patient-sorted PI16⁺ RC cultures (Reviewer Figure 2). Furthermore, in data presented in the accompanying manuscript by De Martin et al., TGFβ3 was found to induce the production of the predicted PI16⁺ RC niche factor, IL-6. These data are shown for the reviewer's attention in Reviewer Figure 2. Together, these data provide mechanistic evidence for TGFβ to induce PI16⁺ RC subset-specification and function.

Reviewer Figure 2. TGFβ induces the differentiation and activation of PI16⁺ RCs. a) Quantification of the relative fold expression of *Pi16* by murine lymph node fibroblasts stimulated with medium or TGFβ. b) Quantification of the relative fold induction of *SEMA3C* transcript by tonsillar FRCs stimulated for 48 hours in vitro with the indicated factors. c) Quantification of the relative fold induction of *SEMA3C* by CD34⁺ PDPN⁺ PI16⁺ RCs sorted from human tonsils and stimulated for 48 hours with TGF-β1.

Lastly, we performed spatial transcriptomics analyses to demonstrate that the inferred BRC-immune cell interactions are topologically plausible. These data are now included in the manuscript in the new Figure 5a-d and Extended Data Figure 5a. Please see our response to Reviewer 2, who had raised a similar concern.

2. The scRNAseq data, as with past data studies by this group, has been deposited as primary data as permitted by the ArrayExpress database. However, the more widely used GEO database requires processed data to be deposited. It can be challenging for labs that lack advanced computing power to download the raw data and run the 10x cellranger pipeline themselves, especially if they are just wanting to quickly query the data. It is recommended that processed data are also deposited. Moreover, many studies that involve largely scRNAseq data acquisition and analysis include a user-friendly portal to facilitate data querying. Such a portal is likely to increase the utility and citation of the study. (It should be noted that the human

palatine tonsil data reported here as available under E-MTAB-11715 is the same dataset analyzed in the accompanying manuscript by De Martin et al., which also indicates depositing the dataset).

We thank the reviewer for this constructive remark. Indeed, the analysis of unprocessed primary data can be challenging and requires advanced computing power. In order to facilitate easy exploration of the processed data, we developed a web-based platform (<https://immbiosg.github.io/FRCdataExplorer/>) for interactive data exploration through an R shiny server. The processed data from this project as well as data from other projects of this group can now be queried via this platform. In addition, we uploaded the processed data files of this project as R objects to figshare (10.6084/m9.figshare.21221291) where they can be downloaded and used for further downstream analysis or exploration in any in-house pipeline after publication of this study. Finally, all code used for data analysis in this project is available at github (https://github.com/mluetge/CrossSLO_BRC_CXCL13). All links to access and explore the data and code are included in the ‘Data availability’ statement.

Reviewer 2:

Pikor and colleagues utilise their Cxcl13-cre-TdTomato strain to characterise CXCL13 expressing cells in lymph nodes, spleen and peyer's patches by scRNAseq and flow cytometry, akin to their 2020 Nat Immunol paper that reported a similar analysis of lymph node CXCL13-expressing cells. By mining these data, they are able to see that there are populations of CXCL13-expressing cells that are common amongst lymphoid organs, including a previously undescribed population of Pi16-expressing stromal cells. In addition, CXCL13 is expressed by many different cell types that are unique to the different organs. The scRNAseq data are used to identify pathways that are enriched in different CXCL13+ stromal cell types. Further scRNAseq of hematopoietic cells followed by interaction analyses identified putative interactions between cell types. A similar scRNAseq analysis was conducted on human lymph nodes, with similar results, highlighting conserved cell types between species.

This manuscript provides a huge array of data, and brings forward many interesting concepts, but none are tested experimentally. The role of Pi16 expressing cells is not clear, only that these can be detected by scRNAseq and imaging. The interactome analysis is interesting, but assuming that this underpins real interactions is incorrect. For example, in figure 4h, plasma cells are predicted to interact with light zone FDCs, however mature plasma cells are not located in B cell follicles, so they are not physically able to interact with FDCs, even though they may have the receptors to do so. Without experimental confirmation of some of the proposed mechanisms/interactions, this paper is descriptive and does not significantly advance our knowledge of B cell interacting reticular cells. The authors have a strong track record in using genetically modified mice to test the role of specific molecules in regulating stromal cell biology. Follow up of interesting pathways and interactions from their scRNAseq data, as they have previously done, would significantly enhance the insight and impact of this paper, and allow the data to support the strong statements in the abstract.

We agree with the reviewer that potential interactions raised by interactome analyses must be carefully considered. For this reason, we, as others in the field are now doing (e.g. Elmentaite et al., 2020 Nature), have identified canonical interactions based on the expression of candidate receptor-ligand transcripts that are conserved across three murine SLOs and two human lymphoid tissues. Still, we agree with the reviewer that the quantification of interactions predicted from transcriptome data can be potentially skewed by false positive inferences if spatial co-localization is not taken into account.

To this end, we generated a new spatial transcriptomics dataset of murine inguinal and mesenteric lymph nodes in order to assess BRC-immune cell crosstalk based on co-localization. Specifically, we utilized the 10X Genomics Visium platform to derive spatially resolved transcriptomes of four biological samples: two inguinal and two mesenteric lymph nodes, each with technical replicates. Following preprocessing, we focused on *Cxcl13*-expressing spots and used spacexr as a tool for robust cell type decomposition based on our single cell RNA-seq data from lymph node BRCs and immune cells. Based on the derived cell type proportions for each spot, we quantified the fraction of shared spots for the conserved BRC subsets with each of the interacting immune cell types. Combined with the interactome analysis, the spatial transcriptomics analysis facilitates the examination of the BRC-immune cell crosstalk in a topologically relevant context. We have now included this higher-level quantification of spatially inferred interactions in the new Figure 5 in the manuscript, replacing the interaction quantification based on single cell data alone.

For the interest of the reviewer, we have provided additional histological validation of a hitherto unknown interaction between plasma cells and PI16⁺ RCs, a newly identified subset of ill-defined function in lymph nodes. Our spatial transcriptomics data demonstrate that plasma cells and PI16⁺ RCs co-localize in regions in the lymph node transected by large vessels, the interfollicular regions at the T-B border as well as in the

medulla (Reviewer Figure 3). The co-localization of plasma cells and BRCs in the interfollicular regions and medulla of the lymph node has been confirmed by high-resolution confocal microscopy (Reviewer Figure 3).

Reviewer Figure 3. Spatial co-localization of PI16⁺ RCs and plasma cells. a) Representative immunofluorescence images of lymph nodes used for Visium spatial transcriptomic analysis stained with the indicated antibodies. b) Localization of *Cxcl13*-expressing BRCs (left) and spatial overlap between PI16⁺ RCs and plasma cells within spots bearing *Cxcl13* transcript (right). c) Confocal microscopy validation of the co-localization of BRCs with plasma cells in the lymph node regions predicted to support the interaction between PI16⁺ RCs and plasma cells. Green colour indicates points of cell-cell interaction.

Moreover, we have employed several complementary in vitro and in vivo approaches to validate the novel interactions sustaining BRC subset specification and immune cell function. These experiments and data are outlined in detail in the responses to Reviewers 1 and 3, who had raised similar concerns. In brief, using in vitro and in vivo approaches, we have demonstrated the role for several of the inferred immune cell-derived factors to promote BRC differentiation and activation. TGF β , Progranulin, VEGF-B and IL4 were found to induce the expression of the BRC-defining chemokine CXCL13. Furthermore, IL-1 β was found to induce FDC expansion, whereas IL-4 promoted FDC function as indicated by increased expression of the niche factor ICAM and enhanced immune complex retention. In terms of PI16⁺ RCs, TGF β was observed to increase *PII6* expression and IL-6 production by sorted CD34⁺ PDPN⁺ RCs from human tonsillar tissues. These data are now included in the manuscript in the new Figure 5 and Extended Data Figures 5 and 8.

To genetically validate the predicted interaction between BRCs and Tfh via IL-6, we assessed GC T and B cell responses in *Cxcl13-Cre/TdTomato Il6^{fl/fl}* mice. BRC-targeted ablation of *Il6* resulted in increased numbers of FoxP3⁺ regulatory follicular helper T cells and attenuated germinal center B cell responses following VSV infection. These data are now presented in the revised manuscript in the new Figure 5. Please see our response to Reviewer 3, who had raised a similar concern.

Reviewer 3:

The manuscript by Lütge et al investigated CXCL13-expressing reticular cell subsets in different secondary lymphoid organs, as well as by comparing samples from mice to humans. Overall findings are that B cell-interacting reticular cell (BRC) subsets shared similarities across different SLOs in terms of gene expression and topology, but there were tissue-specific gene signatures for each subset. Shared BRC subsets in different SLOs had canonical subset markers, as well as gene expression programs unique to each SLO, based on GO term analysis. Conclusions are drawn about BRC-immune cell interactions and that BRCs govern humoral immune responses.

1. The authors make a number of broad mechanistic conclusions but do not undertake further experimental exploration of testable hypotheses. For example: subset-specific genes – can you modulate tissue-specific subset identity or function by modulating gene expression? Will it change the outcome of local responses? Furthermore, the language used in the conclusions is not completely supported by the data at hand and would be more appropriate when mechanistic experiments are performed (eg ‘no functional overlap’, ‘bidirectional molecular circuits’, ‘feedforward circuits’, ‘...suggestive of conserved BRC subset-specific interactions steering regionalized niches in and around the B cell follicle’.)

We thank the reviewer for this comment. As detailed in the responses to Reviewer 1, we have now performed extensive validation experiments to demonstrate the biological function of the newly inferred factors that affect BRC activation and differentiation. In vitro stimulation of murine lymph node fibroblasts has highlighted a role for the BRC differentiation/activation cues (TGF β , Progranulin, VEGF-B and IL-4) to induce the expression of the BRC-defining chemokine CXCL13. In human tonsillar fibroblast cultures, IL-1 β was found to induce the expression of the FDC-associated transcription factor *SOX9* and to stimulate the expansion of FDCs in the murine in vivo setting. In turn, IL-4 was found to induce the expression of genes encoding the activation markers ICAM and SCA-1, as well as promote immune complex retention by FDCs. Lastly, in human tonsillar FRC cultures, TGF β induced *PI16* expression and the production of IL-6, a predicted PI16⁺ RC niche factor. These data are now provided in the new Figure 5 and Extended Data Figures 5 and 8.

The reviewer raises several concerns regarding the choice of language, highlighting gaps in clarity based on the data-intensive bioinformatics approach used in the study without additional validation. The additional validation experiments requested by all reviewers have greatly increased clarity in terms of the biological function of these bidirectional interactions. Indeed, we are now able to distinguish between BRC differentiation and activation, with the former reflecting more subset specification, and activation being a feature that further aids in the ability of BRCs to orchestrate immune responses (eg., the ability of IL-4 to increase adhesion molecules and immune complex retention). We have now replaced the wording ‘maturation’ with differentiation or activation in the manuscript.

We also thank the reviewer for highlighting a lack of clarity regarding the subset-specific biological effects. Indeed, the outcome of immune cell-derived cytokines acts on BRCs to induce a unique biological outcome in each BRC subset, but not necessarily acting exclusively on one subset. We acknowledge that summarizing conserved interaction pairs in a schematic could allow the reader to conclude that the immune cell derived factors act uniquely on one BRC subset via uniquely the one receptor. In line with the key conclusions of the study – that immune cells shape BRC niches to enhance humoral immunity across organs and species – we were able to experimentally validate the differentiating or activating function of immune cell-derived cues on BRCs, rather than directly show that the effect is mediated via a given receptor. We have now simplified this summary schematic to show the directionality of validated immune cell derived cues acting on FDCs and PI16⁺ RCs, and the feedforward effects in terms of niche factor production. We

were also able to validate the immunological impact of one such niche factor, IL6, using genetic ablation in *Cxcl13*-Cre targeted cells (detailed below).

In terms of conceptual terminology, our bioinformatics-based deduction and experimental validation indeed substantiate our claims of bidirectional cross-talk, with immune cell-derived factors having the potential to steer BRC differentiation and activation. We have however, updated the wording of ‘functional overlap’ to reflect overlap in terms of BRC-derived niche factors (lines 70 – 71). Further use of terms of such as conserved interactions or regionalized niches refers to the reiteration of the receptor-ligand pairs between BRCs and immune cells across three murine and two human lymphoid tissues, and the spatial localization of such interactions via histology and spatial transcriptomics. We feel that this language is important for the conceptualization and accessibility of such data-intensive transcriptomic analyses to a broader audience.

2. BRC-immune cell interactions were predicted from droplet-based scRNAseq and the significance of these interactions in terms of humoral immunity was inferred. This is another area where further mechanistic insight could be generated. For example, the authors infer improved B cell immunity based on enrichment of specific signalling pathways. If inferring role of BRCs in modulating GC responses, this be demonstrated with conditional KO studies. If BRCs don't express these receptors are GC responses altered or abrogated? Is the 'efficacy of humoral immunity' altered when these pathways are dysregulated?

We thank the reviewer for this comment. Our study identifies several conserved interactions that encompass BRC-derived niche factors with well-defined receptors on immune cells. We had inferred functional consequences on humoral immunity based on prior knowledge of the biological function of these receptors (eg, the role of BAFF-R, IL-7R, CXCR5, CCR7 in steering lymphocyte survival and recruitment). In addition to these known niche factors, our stringent bioinformatics approach has identified hitherto unknown niche factors and interactions, including the IL-6-mediated crosstalk between *Pi16*⁺ RCs and follicular helper T cells (Tfh). To demonstrate the functional relevance of BRC-derived IL-6 in terms of humoral immunity, we have assessed antiviral responses at the peak of the GC response in *Cxcl13*-Cre/TdTomato *Il6*^{fl/fl} mice. Genetic ablation of *Il6* was found to affect Tfh differentiation, promoting a regulatory follicular helper phenotype that culminated in a decreased frequency of GC B cells and reduced systemic neutralizing antibody titers following viral infection. This data is now included in Extended Data Figure 6. Experimental dissection of the many additional BRC differentiation and activation factors will be followed in subsequent studies.

As we were intrigued as to where *PI16*⁺ RCs and Tfh could interact, we examined the co-localization of each population in our new spatial transcriptomics dataset. These interactions indeed were observed to occur in interfollicular and subcapsular regions and the medulla. This likely reflects an underappreciated mobility of T cells. These interactions have been visualized for the attention of the reviewer (Reviewer Figure 4).

Reviewer Figure 4. Spatial co-localization of Tfh and *PI16*⁺ RCs. a) Visualization of *Cxcl13* expressing BRCs in murine lymph nodes using the Visium spatial transcriptomic platform. **b)** Visualization of co-localization of *PI16*⁺ RCs and Tfh in lymph nodes.

Minor comments

There were a few sentences that weren't clear in its intent or need references, eg

1) Lines 54-57 appears to be conflating germinal centers with BRCs

We thank the reviewer for noting this. We have amended the text.

2) When discussing 'DZ FDCs' – can the authors please provide references.

We have added the appropriate references, e.g. Pikor et al. 2020.

Decision Letter, first revision:

12th Dec 2022

Dear Burkhard,

We have now finished reviewing your revised manuscript entitled "Conserved stromal-immune cell circuits secure B cell homeostasis and function", reference number NI-A34044A. We obtained 2 of the original 3 referees to have another look at the study. While both felt the manuscript had been improved, referee #2 was more critical of this particular study looking specifically at the CXCL13+ stromal cell population. In their opinion, this study remains too descriptive and that the stromal-seq analysis used shows spatial proximity but it does not have sufficient resolution to allow conclusions about direct cell-cell communication. Hence, an unambiguous endorsement for publication in Nature Immunology was not offered by this referee.

[REDACTED]

I hope that you continue to consider Nature Immunology for your results most significant for the immunology community and wish you well in your future investigations.

Kind regards,

Laurie

Laurie A. Dempsey, Ph.D.
Senior Editor
Nature Immunology
l.dempsey@us.nature.com
ORCID: 0000-0002-3304-796X

Reviewers' comments:

Reviewer #2 (Remarks to the Author):

The authors have added new data in response to my concerns about there being a lack of experimental confirmation of observations from the scRNAseq data analysis. Whilst these new data provide more functional insight than was provided previously, it does not confirm any of the putative interactions, nor provide insight into the role of Pi16 expressing stromal cells. Whilst these data are of interest, they still do not support the strong statements/conclusions in the abstract. The major issue with the paper is the disconnect between the conclusions and the data.

In the revised manuscript the authors add 10x visium spatial transcriptomics and perform some experiments to extend their interactome analysis. The spatial transcriptomics provides proximity mapping, but does not provide single cell resolution, so true cell-cell interactions cannot be mapped. This limitation needs to be clearly stated and the authors interpretation of the data adjusted accordingly.

The functional relevance of three cytokines was tested by administration in vivo, and this demonstrated that FDCs respond to IL-1b and IL-4 (consistent with PMID: 34555336). These are interesting functional data, but do not confirm a cell-cell interaction.

To determine if there is a functional role for IL-6 production by Cxcl13 expressing cells or their daughters the Il6fl/fl Cxcl13cre mice were generated. Whilst Tfh cell numbers were not affected, there was an expansion of T follicular regulatory cells, and a modest reduction in germinal center B cells and anti-VSV antibodies. This is an interesting and novel finding, but does not confirm a predicted interaction from the interactome analysis.

Figure 5s needs to be removed, as the data does not support the proposed interactions.

Reviewer #3 (Remarks to the Author):

The authors have satisfactorily addressed my comments with their addition of new data.

[REDACTED]

Decision Letter, Appeal

Dear Burkhard,

Thank you for supplying your rebuttal to the referees comments and to our editorial concerns to your manuscript entitled "Conserved stromal-immune cell circuits secure B cell homeostasis and function". Please revise your manuscript as we discussed previously in our email correspondence.

We therefore invite you to revise your manuscript taking into account all reviewer and editor comments. Please highlight all changes in the manuscript text file in Microsoft Word format.

Once you have made these revisions, please use the URL below to submit the revised manuscript with figures, an updated life science reporting summary and any supplemental checklists, and a point-by-point response addressing the reviewers' criticisms.

The Reporting Summary can be found here:
<https://www.nature.com/documents/nr-reporting-summary.pdf>

The Editorial Policy Checklist can be found here: <https://www.nature.com/documents/nr-editorial-policy-checklist.pdf>

[REDACTED]

Please let us know how you wish to proceed and when we can expect your revised manuscript.

With kind regards,

Laurie

Laurie A. Dempsey, Ph.D.
Senior Editor
Nature Immunology
l.dempsey@us.nature.com
ORCID: 0000-0002-3304-796X

Author Rebuttal, first revision:

See Inserted PDF

Dear Laurie, Dear Editors of Nature Immunology,

We would like to thank the Editors for the opportunity to revise our manuscript entitled “Conserved stromal-immune cell circuits secure B cell homeostasis and function”. We are pleased that Reviewer 3 found that we had satisfied all of their comments with the added experiments that validated the predicted effect of immune cell-provided factors on BRC identity and specification. Our revised manuscript addresses the Editor’s and Reviewer 2’s outstanding comments, the latter which are addressed in a point-by-point reply below.

Reviewer #2 (Remarks to the Author):

The authors have added new data in response to my concerns about there being a lack of experimental confirmation of observations from the scRNAseq data analysis. Whilst these new data provide more functional insight than was provided previously, it does not confirm any of the putative interactions, nor provide insight into the role of Pi16 expressing stromal cells. Whilst these data are of interest, they still do not support the strong statements/conclusions in the abstract. The major issue with the paper is the disconnect between the conclusions and the data

In the revised manuscript the authors add 10x visium spatial transcriptomics and perform some experiments to extend their interactome analysis. The spatial transcriptomics provides proximity mapping, but does not provide single cell resolution, so true cell-cell interactions cannot be mapped. This limitation needs to be clearly stated and the authors interpretation of the data adjusted accordingly.

We appreciate the Reviewer’s comments regarding the lack of direct evidence showing cell-cell interactions. We feel confident that the thoroughness and consistency of the transcriptional and experimental data defining: 1) conserved BRC niche factor and signalling profiles; 2) predicted immune cell-provided ligands; 3) the biological effect of cues provided by immune cells on BRC differentiation and activation in both mouse and human lymphoid organs – substantiates a true biological cross-talk between immune cells and BRCs. We recognize that as we have not shown direct cell-cell contact, we should adjust our wording regarding BRC-immune cell interactions. Throughout the revised manuscript we have atoned the wording to reflect an impact of soluble, immune cell-provided cues on BRC differentiation and activation. These changes are reflected in the abstract and throughout the revised manuscript in lines: 28-30, 75, 243, 410, 453, 463-464, 475, 499, 501-502, 506, 510, 515-516, 526, 551, 554, 559, 574.

It is important to stress that interactome analyses and spatial transcriptomics have been used to reveal potential ligands driving convergent BRC subset specification, and infer proximity of immune cells that are the dominant expressers of BRC specification cues with predicted BRC receiver subsets, respectively. Additionally, mapping BRC subset-specific signalling gene signatures to the appropriate BRC niche was particularly useful to map PI16⁺ RCs, as we could not succeed to stain for this peptidase inhibitor in murine tissues.

We agree with the reviewer that the 10x Visium platform does not provide single cell resolution. This limitation is now stated in the revised manuscript at lines 300-302 and the interpretation of the data was adjusted to highlight that our findings are based on careful bioinformatic deduction (lines 315, 316). Nevertheless, the lack of resolution from spatial transcriptomics does not nullify the performed in vitro and in vivo validation experiments, as:

- all immune cell-derived factors predicted to act on BRCs are soluble, thus could have an effect within a 27.5 µm radius – the size of one spot on the 10X chip (soluble factors in the B cell follicle are predicted to act within a 50 µm field, PMID: 32699279), and

- each immune cell is in direct physical contact with stromal cells in the densely-packed B cell follicle.

For the interest of the Reviewer, we have provided histological images of direct cell-cell contact between lymphocytes and PI16⁺ RCs or FDCs in the human tonsil or mouse lymph nodes, respectively.

Reviewer Figure 1: FRC-immune cell contact in secondary lymphoid organs. a,b) High-resolution reconstruction of the adult subepithelial niche in human palatine tonsil showing surface contact areas (blue) of FBLN1⁺ cells with CD20⁺ (arrows) and CD3⁺ (arrowheads) lymphocytes. FBLN1 was used as surrogate marker for PI16⁺ RCs. Images were taken from the accompanying manuscript by De Martin et al. (patient 1) for the attention of the reviewer. **c,d)** High-resolution reconstruction of a murine lymph node B cell follicle visualizing surface contact areas (purple) between Cxcl13-Cre/TdTom⁺CR2⁺ FDCs with PD1⁺ Tfh cells.

The functional relevance of three cytokines was tested by administration in vivo, and this demonstrated that FDCs respond to IL-1b and IL-4 (consistent with PMID: 34555336). These are interesting functional data, but do not confirm a cell-cell interaction.

Our in vitro and in vivo stimulation experiments validate the biological effect of five (5) predicted immune cell-derived BRC specification cues: IL-4, IL-1 β , VEGF-B, TGF- β 1 and PGRN (Figure 5e-l). We are grateful to the Reviewers for requesting these validation experiments, which have elucidated the hitherto unknown biological impact of the predicted immune cell-provided ligands (all of which are soluble) on BRC differentiation or activation (Figure 5e-l).

The Reviewer cites recent work from Jason Cyster's group demonstrating a role for FDCs to sequester IL-4 to promote T cell activation (PMID: 34555336). We agree that T cell-related functions of IL-4 binding to its receptor on FDCs is of interest and have already cited this work in the discussion. However, Duan et al. have not examined the FDC-intrinsic role of IL-4R signalling, leaving this axis specifying FDC activation by immune cell-provided factors hitherto unknown. In revealing IL-4 as an important FDC maturation cue, our study opens further questions pertaining to the differential or synergistic role of cytokine signalling in FDCs and T cells on GC outcome.

To determine if there is a functional role for IL-6 production by Cxcl13 expressing cells or their daughters the Il6fl/fl Cxcl13cre mice were generated. Whilst Tfh cell numbers were not affected, there was an expansion of T follicular regulatory cells, and a modest reduction in germinal center B cells and anti-VSV antibodies. This is an interesting and novel finding, but does not confirm a predicted interaction from the interactome analysis.

We are pleased to see that the Reviewer acknowledges the novelty of the observed role for BRC-provided IL-6 to steer follicular regulatory T cell differentiation. Indeed, we cannot conclude with these genetic experiments alone that PI16⁺ RC-derived IL-6 skews Tfh differentiation. For this reason, we have been careful in the text to refer to the role of Cxcl13-Cre-provided IL-6. However, considering the below-listed insight from our single cell and spatial transcriptomic data, we can infer that PI16⁺ RCs are a relevant Cxcl13-Cre-targeted, *Il6*-expressing subset that affects Tfh differentiation to sustain GC responses.

- PI16⁺ RCs are the dominant expressers of *Il6* across murine and human SLOs (Figure 4j, and Extended Data Figures 4d, 9d)
- Tfh are a key cell type expressing the IL-6 receptor (Figures 4j and 7j)
- proximal localization of *Pi16*⁺ RCs and Tfh in all biological and technical spatial transcriptomic replicates (Figure 5d)

Figure 5s needs to be removed, as the data does not support the proposed interactions.

The schematic in Figure 5m (previously Figure 5s) visualizes the immune cell-provided ligands that were predicted to signal to the respective BRC subset and were validated to have an effect on BRC differentiation or maturation (as measured by the production of ICAM, CR2 or IL-6). Not only does our interactome analysis show that T and B cells express the receptors for these BRC niche factors (ie., *Itgal + Itgb2, Fcgr2a, Il6ra*), but the biological relevance of these niche factors is either known or demonstrated in our study in the case of IL-6. For instance, the expression of complement receptors by FDCs is well accepted to substantiate antigen sampling by B cells, and ICAM1 has been shown to support Tfh function (PMID 27760339).

We accept that our in vitro and in vivo validation experiments do not formally prove that BRC specification cues (IL-4, IL-1 β , VEGF-B, TGF- β 1 and PGRN) are produced by the indicated B or T lymphocytes, or myeloid cell populations. This would require cell type-specific gene deletion experiments that are beyond the scope of the current study. However, we feel that this schematic is an important visualization of a main take-home of the paper (ie. that conserved, bidirectional signalling programs sustain BRC niches), consolidating the transcriptomic analysis and experimental validations in Figures 4 and 5. To this end, we have atoned the wording (lines 342-347) to stress that the schematic represents inferred interactions based on expression data and in vitro / in vivo stimulation experiments.

Reviewer #3 (Remarks to the Author):

The authors have satisfactorily addressed my comments with their addition of new data.

Decision Letter, second revision:

Our ref: NI-A34044C

9th Mar 2023

Dear Burkhard,

Thank you for submitting your revised manuscript "Conserved stromal-immune cell circuits secure B cell homeostasis and function" (NI-A34044C). As with the companion manuscript, NI-A34043C, I've looked over the current revisions and we are satisfied that they have addressed the remaining concerns posed by referee #2. Therefore we'll be happy in principle to publish it in Nature Immunology, pending minor revisions to comply with our editorial and formatting guidelines.

We will now perform detailed checks on your paper and will send you a checklist detailing our editorial and formatting requirements in about a week. Please do not upload the final materials and make any revisions until you receive this additional information from us.

If you had not uploaded a Word file for the current version of the manuscript, we will need one before beginning the editing process; please email that to immunology@us.nature.com at your earliest convenience.

Thank you again for your interest in Nature Immunology Please do not hesitate to contact me if you have any questions. {Please note, however, that I'll be traveling away from the office for the next 10 days - as I'll be in Taiwan for conference & lab visits - hence I might not be able to respond right away.}

Kind regards,

Laurie

Laurie A. Dempsey, Ph.D.
Senior Editor
Nature Immunology
l.dempsey@us.nature.com
ORCID: 0000-0002-3304-796X

Final Decision Letter:

Dear Burkard & Natalia,

I am delighted to accept your manuscript entitled "Conserved stromal-immune cell circuits secure B cell homeostasis and function" for publication in an upcoming issue of Nature Immunology.

Over the next few weeks, your paper will be copyedited to ensure that it conforms to Nature Immunology style. Once your paper is typeset, you will receive an email with a link to choose the appropriate publishing options for your paper and our Author Services team will be in touch regarding

any additional information that may be required.

Please note that *Nature Immunology* is a Transformative Journal (TJ). Authors may publish their research with us through the traditional subscription access route or make their paper immediately open access through payment of an article-processing charge (APC). Authors will not be required to make a final decision about access to their article until it has been accepted. [Find out more about Transformative Journals](https://www.springernature.com/gp/open-research/transformative-journals).

Authors may need to take specific actions to achieve [compliance with funder and institutional open access mandates](https://www.springernature.com/gp/open-research/funding/policy-compliance-faqs). If your research is supported by a funder that requires immediate open access (e.g. according to [Plan S principles](https://www.springernature.com/gp/open-research/plan-s-compliance)) then you should select the gold OA route, and we will direct you to the compliant route where possible. For authors selecting the subscription publication route, the journal's standard licensing terms will need to be accepted, including [self-archiving policies](https://www.springernature.com/gp/open-research/policies/journal-policies). Those licensing terms will supersede any other terms that the author or any third party may assert apply to any version of the manuscript.

Your paper will be published online soon after we receive your corrections and will appear in print in the next available issue. Content is published online weekly on Mondays and Thursdays, and the embargo is set at 16:00 London time (GMT)/11:00 am US Eastern time (EST) on the day of publication. Now is the time to inform your Public Relations or Press Office about your paper, as they might be interested in promoting its publication. This will allow them time to prepare an accurate and satisfactory press release. Include your manuscript tracking number (NI-A34044D) and the name of the journal, which they will need when they contact our office.

About one week before your paper is published online, we shall be distributing a press release to news

organizations worldwide, which may very well include details of your work. We are happy for your institution or funding agency to prepare its own press release, but it must mention the embargo date and Nature Immunology. Our Press Office will contact you closer to the time of publication, but if you or your Press Office have any enquiries in the meantime, please contact press@nature.com.

Also, if you have any spectacular or outstanding figures or graphics associated with your manuscript - though not necessarily included with your submission - we'd be delighted to consider them as candidates for our cover. Simply send an electronic version (accompanied by a hard copy) to us with a possible cover caption enclosed.

If you have not already done so, we strongly recommend that you upload the step-by-step protocols used in this manuscript to the Protocol Exchange. Protocol Exchange is an open online resource that allows researchers to share their detailed experimental know-how. All uploaded protocols are made freely available, assigned DOIs for ease of citation and fully searchable through nature.com. Protocols can be linked to any publications in which they are used and will be linked to from your article. You can also establish a dedicated page to collect all your lab Protocols. By uploading your Protocols to Protocol Exchange, you are enabling researchers to more readily reproduce or adapt the methodology you use, as well as increasing the visibility of your protocols and papers. Upload your Protocols at www.nature.com/protocolexchange/. Further information can be found at www.nature.com/protocolexchange/about .

Please note that we encourage the authors to self-archive their manuscript (the accepted version before copy editing) in their institutional repository, and in their funders' archives, six months after publication. Nature Portfolio recognizes the efforts of funding bodies to increase access of the research they fund, and strongly encourages authors to participate in such efforts. For information about our editorial policy, including license agreement and author copyright, please visit www.nature.com/ni/about/ed_policies/index.html

Kind regards,

Laurie

Laurie A. Dempsey, Ph.D.
Senior Editor
Nature Immunology
l.dempsey@us.nature.com
ORCID: 0000-0002-3304-796X